# KSD Aggregated Goodness-of-fit Test

**Antonin Schrab**
Centre for Artificial Intelligence
Gatsby Computational Neuroscience Unit
University College London & Inria London
a.schrab@ucl.ac.uk

**Benjamin Guedj**
Centre for Artificial Intelligence
University College London & Inria London
b.guedj@ucl.ac.uk

**Arthur Gretton**
Gatsby Computational Neuroscience Unit
University College London
arthur.gretton@gmail.com

## Abstract

We investigate properties of goodness-of-fit tests based on the Kernel Stein Discrepancy (KSD). We introduce a strategy to construct a test, called KSDAGG, which aggregates multiple tests with different kernels. KSDAGG avoids splitting the data to perform kernel selection (which leads to a loss in test power), and rather maximises the test power over a collection of kernels. We provide non-asymptotic guarantees on the power of KSDAGG: we show it achieves the smallest uniform separation rate of the collection, up to a logarithmic term. For compactly supported densities with bounded model score function, we derive the rate for KSDAGG over restricted Sobolev balls; this rate corresponds to the minimax optimal rate over unrestricted Sobolev balls, up to an iterated logarithmic term. KSDAGG can be computed exactly in practice as it relies either on a parametric bootstrap or on a wild bootstrap to estimate the quantiles and the level corrections. In particular, for the crucial choice of bandwidth of a fixed kernel, it avoids resorting to arbitrary heuristics (such as median or standard deviation) or to data splitting. We find on both synthetic and real-world data that KSDAGG outperforms other state-of-the-art quadratic-time adaptive KSD-based goodness-of-fit testing procedures.

## 1 Introduction

Kernel selection remains a fundamental problem in kernel-based nonparametric hypothesis testing, as it significantly impacts the test power. Kernel selection has attracted a significant interest in the literature, and a number of methods have been proposed in the two-sample, independence and goodness-of-fit testing frameworks, such as using heuristics (Gretton et al., 2012a), relying on data splitting (Gretton et al., 2012b; Sutherland et al., 2017; Kübler et al., 2022), learning deep kernels (Grathwohl et al., 2020; Liu et al., 2020), working in the post-selection inference framework (Yamada et al., 2019; Lim et al., 2019, 2020; Kübler et al., 2020; Freidling et al., 2021), to name but a few.

In this work, we focus on aggregated tests, which have been investigated for the two-sample problem by Fromont et al. (2013), Kim et al. (2022) and Schrab et al. (2021) using the Maximum Mean Discrepancy (MMD; Gretton et al., 2012a), and for the independence problem by Albert et al. (2022) and Kim et al. (2022) using the Hilbert Schmidt Independence Criterion (HSIC; Gretton et al., 2005). We extend the use of aggregated tests to the goodness-of-fit setting, where we are given a model and some samples, and test whether the samples have been drawn from the model. We employ the Kernel Stein Discrepancy (KSD; Chwialkowski et al., 2016; Liu et al., 2016) as our test statistic, which is an ideal measure of distance for this setting: it admits an estimator which can be computed without

requiring samples from the model, and does not require the model to be normalised. To the best of our knowledge, ours represents the first aggregation procedure for the KSD test in the literature.

**Related work.** Fromont et al. (2012, 2013) introduced non-asymptotic aggregated tests for the two-sample problem with sample sizes following a Poisson process, using an unscaled version of the MMD. Using a wild bootstrap, they derived uniform separation rates (Ingster, 1987, 1989, 1993a,b; Baraud, 2002). Albert et al. (2022) then proposed an independence aggregated test using the HSIC, with guarantees using a theoretical quantile, but relying on permutations to obtain the test threshold in practice. Kim et al. (2022) then extended those theoretical results to also hold for the quantile estimated using permutations, however, they did not obtain the desired level dependency in their HSIC bound. All those aforementioned results were proved for the Gaussian kernel only. Schrab et al. (2021) generalised the two-sample results to hold for the usual MMD estimator and for a wide range of kernels using either a wild bootstrap or permutations, and provided optimality results which hold with fewer restrictions. Our work builds and extends on the above non-asymptotic results: we consider the goodness-of-fit framework, where we have samples from only one of the two densities. The main challenges arise from working with the Stein kernel which defines the KSD test statistic: for example, we lose the transition-invariant property of the kernel which is crucial to work in the Fourier domain. Balasubramanian et al. (2021) considered adaptive MMD-based goodness-of-fit tests and obtained their uniform separation rates over Sobolev balls in the asymptotic regime. More generally, Li and Yuan (2019) studied asymptotic adaptive kernel-based tests for the three testing frameworks. Tolstikhin et al. (2016) derived minimax rates for MMD estimators using radial universal kernel (Sriperumbudur et al., 2011). Schrab et al. (2022) extends this work, together with those of Albert et al. (2022) and Schrab et al. (2021), to construct efficient aggregated tests for the three testing frameworks using incomplete $U$-statistics. They quantify the cost in the minimax rate over Sobolev balls incurred for computational efficiency, and prove minimax optimality of the quadratic-time HSIC permuted aggregated test by improving the bound of Kim et al. (2022). See Appendix C for details.

**Contributions.** We propose a solution to the fundamental kernel selection problem for the widely-used KSD goodness-of-fit tests: we construct an adaptive test KSDAGG which aggregates multiple tests with different kernels. Our contribution is in showing, both theoretically and experimentally, that the aggregation procedure works in this novel setting in which it has never been considered before. We work in the kernel selection framework; this general setting has many applications including the one of kernel bandwidth selection. Our aggregated test allows for two numerical methods for estimating the test thresholds: the wild bootstrap and the parametric bootstrap (a procedure unique to the goodness-of-fit framework). We conduct a theoretical analysis: we derive a general condition which guarantees test power for KSDAGG in terms of its uniform separation rate, with extra assumptions including regularity over restricted Sobolev balls we prove that KSDAGG attains the minimax rate over (unrestricted) Sobolev balls. We discuss the implementation of KSDAGG and experimentally validate our proposed approach on benchmark problems, not only on datasets classically used in the literature but also on original data obtained using state-of-the-art generative models (*i.e.* Normalizing Flows). We observe, both on synthetic and real-world data, that KSDAGG obtains higher power than other KSD-based adaptive state-of-the-art tests. Contributing to the real-world applications of these goodness-of-fit tests, we provide publicly available code to allow practitioners to employ our method: `https://github.com/antoninschrab/ksdagg-paper`.

## 2 Notation

We consider the goodness-of-fit problem where given access to a known model probability density $p$ (which can be unnormalised since we actually only need access to its score function $\nabla \log p(\cdot)$) and to some i.i.d. $d$-dimensional samples $\mathbb{X}_N := (X_i)_{i=1}^N$ drawn from an unknown data density $q$, we want to decide whether $p \neq q$ holds. This can be expressed as a statistical hypothesis testing problem with null hypothesis $\mathcal{H}_0 : p = q$ and alternative $\mathcal{H}_a : p \neq q$.

As a measure of distance between $p$ and $q$, we use the *Kernel Stein Discrepancy* (KSD) introduced by Chwialkowski et al. (2016) and Liu et al. (2016). For a kernel $k$, the KSD is the Maximum Mean Discrepancy (MMD; Gretton et al., 2012a) between $p$ and $q$ using the Stein kernel associated to $k$

$$\text{KSD}_{p,k}^2(q) := \text{MMD}_{h_{p,k}}^2(p,q) := \mathbb{E}_{q,q}[h_{p,k}(X,Y)] - 2\mathbb{E}_{p,q}[h_{p,k}(X,Y)] + \mathbb{E}_{p,p}[h_{p,k}(X,Y)]$$
$$= \mathbb{E}_{q,q}[h_{p,k}(X,Y)]$$

where the *Stein kernel* $h_{p,k} \colon \mathbb{R}^d \times \mathbb{R}^d \to \mathbb{R}$ is defined as

$$h_{p,k}(x,y) \coloneqq \left(\nabla \log p(x)^\top \nabla \log p(y)\right) k(x,y) + \nabla \log p(y)^\top \nabla_x k(x,y)$$

$$+ \nabla \log p(x)^\top \nabla_y k(x,y) + \sum_{i=1}^{d} \frac{\partial}{\partial x_i \partial y_i} k(x,y)$$

and satisfies the *Stein identity* $\mathbb{E}_p[h_{p,k}(X,\cdot)] = 0$. Additional background details on the KSD are presented in Appendix C. A quadratic-time *KSD estimator* can be computed as the $U$-statistic (Hoeffding, 1992)

$$\widehat{\mathrm{KSD}}_{p,k}^2(\mathbb{X}_N) \coloneqq \frac{1}{N(N-1)} \sum_{1 \le i \ne j \le N} h_{p,k}(X_i, X_j). \tag{1}$$

In this work, the score of the model density $p$ is always known, we do not always explicitly write the dependence on $p$ of all the variables we consider. We assume that the kernel $k$ is such that

$$\mathrm{KSD}_{p,k}^2(q) = \mathbb{E}_{q,q}[h_{p,k}(X,Y)] < \infty \qquad \text{and} \qquad C_k \coloneqq \mathbb{E}_{q,q}[h_{p,k}(X,Y)^2] < \infty. \tag{2}$$

We now address the requirements for consistency (*i.e.* test power converges to 1 as the sample size goes to $\infty$) of the Stein test (Chwialkowski et al., 2016, Theorem 2.2): we assume that the kernel $k$ is $C_0$-universal (Carmeli et al., 2010, Definition 4.1) and that $\mathbb{E}_q\left[\left\|\nabla\left(\log \frac{p(X)}{q(X)}\right)\right\|_2^2\right] < \infty$.

We use the notations $\mathbb{P}_p$ and $\mathbb{P}_q$ to denote the probability under the model distribution $p$ and under the data distribution $q$, respectively. Given a kernel $\kappa \colon \mathbb{R}^d \times \mathbb{R}^d \to \mathbb{R}$ and a function $f \colon \mathbb{R}^d \to \mathbb{R}$ in $L^2(\mathbb{R}^d)$, we consider the *integral transform* $T_\kappa$ defined as

$$(T_\kappa f)(y) \coloneqq \int_{\mathbb{R}^d} \kappa(x,y) f(x) \, \mathrm{d}x$$

for $y \in \mathbb{R}^d$. When the kernel $\kappa$ is translation-invariant, the integral transform corresponds to a convolution. However, the lack of translation invariance of the Stein kernel introduces new challenging problems. First, working the integral transform of the Stein kernel is more complicated since it does not correspond to a simple convolution. Second, for the expectation of the Stein kernel squared, $C_k \coloneqq \mathbb{E}_{q,q}[h_{p,k}(X,Y)^2]$, it is not possible to extract the bandwidth parameter $\lambda$ outside of the expectation as it is the case when using the usual kernel directly as for the MMD and HSIC.

## 3 Construction of tests and bounds

We now introduce the single and aggregated KSD tests. We show that these control the probability of type I error as desired, and provide conditions for the control of the probability of type II error.

### 3.1 Single test

We first construct a KSD test for a fixed kernel $k$ as proposed by Chwialkowski et al. (2016) and Liu et al. (2016). To estimate the test threshold, we can either use a wild bootstrap (Shao, 2010; Leucht and Neumann, 2013; Fromont et al., 2012; Chwialkowski et al., 2014) or a parametric bootstrap (Stute et al., 1993). Both methods work by simulating sampling values $(\bar{K}_k^1, \ldots, \bar{K}_k^{B_1})$ from the (asymptotic) distribution of $\widehat{\mathrm{KSD}}_{p,k}^2$ under the null hypothesis and estimating the $(1-\alpha)$-quantile for $\alpha \in (0,1)$ using a Monte Carlo approximation[1]

$$\widehat{q}_{1-\alpha}^k \coloneqq \inf\left\{u \in \mathbb{R} : 1 - \alpha \le \frac{1}{B_1 + 1} \sum_{b=1}^{B_1+1} \mathbb{1}\left(\bar{K}_k^b \le u\right)\right\} = \bar{K}_k^{\bullet\lceil (B_1+1)(1-\alpha)\rceil} \tag{3}$$

where $\bar{K}_k^{\bullet 1} \le \cdots \le \bar{K}_k^{\bullet B_1+1}$ are the sorted elements $(\bar{K}_k^1, \ldots, \bar{K}_k^{B_1+1})$ with $\bar{K}_k^{B_1+1} \coloneqq \widehat{\mathrm{KSD}}_{p,k}^2(\mathbb{X}_N)$. The single test is then defined as (a test function outputs 1 when the null is rejected and 0 otherwise)

$$\Delta_\alpha^k(\mathbb{X}_N) \coloneqq \mathbb{1}\left(\widehat{\mathrm{KSD}}_{p,k}^2(\mathbb{X}_N) > \widehat{q}_{1-\alpha}^k\right).$$

---

[1]We do not write explicitly the dependence of $\widehat{q}_{1-\alpha}^k$ on other variables, but those are implicitly considered when writing probabilistic statements.

For the *parametric bootstrap*, we directly draw new samples $(X_i')_{i=1}^N$ from the model distribution $p$ (it might not always be possible to so) and compute the KSD

$$\bar{K}_k := \frac{1}{N(N-1)} \sum_{1 \leq i \neq j \leq N} h_{p,k}(X_i', X_j'). \tag{4}$$

For the *wild bootstrap*, we first generate $n$ i.i.d. Rademacher random variables $\epsilon_1, \ldots, \epsilon_n$, each taking value in $\{-1, 1\}$, and then compute

$$\bar{K}_k := \frac{1}{N(N-1)} \sum_{1 \leq i \neq j \leq N} \epsilon_i \epsilon_j h_{p,k}(X_i, X_j). \tag{5}$$

By repeating either procedure $B_1$ times, we obtain the bootstrapped samples $\bar{K}_k^1, \ldots, \bar{K}_k^{B_1}$.

Since it uses samples from the model $p$, the parametric bootstrap (Stute et al., 1993) results in a test with non-asymptotic level $\alpha$. This comes at the cost of being computationally more expensive and assuming that we are able to sample from $p$ (which may be out of reach in some settings). Conversely, the wild bootstrap has the advantage of not requiring to sample from $p$, which makes it computationally more efficient as only one kernel matrix needs to be computed, but it only achieves the desired level $\alpha$ asymptotically (Shao, 2010; Leucht and Neumann, 2013; Leucht, 2012; Chwialkowski et al., 2014, 2016) assuming Lipschitz continuity of $h_{p,k}$ (see Appendix D for details). Note that we cannot obtain a non-asymptotic level for the wild bootstrap by relying on the result of Romano and Wolf (2005, Lemma 1) as done in the two-sample framework by Fromont et al. (2013) and Schrab et al. (2021). This is because in our case $\bar{K}_k$ and $\widehat{\mathrm{KSD}}_{p,k}^2(\mathbb{X}_N)$ are not exchangeable variables under the null hypothesis, due to the asymmetry of the KSD statistic with respect to $p$ and $q$.

Having discussed control of the probability of type I error of the single test $\Delta_\alpha^k$, we now provide a condition on $\|p - q\|_2$ which ensures that the probability of type II error is controlled by some $\beta \in (0,1)$. The smallest such value of $\|p - q\|_2$, provided that $p - q$ lies in some given class of regular functions, is called the *uniform separation rate* (Ingster, 1987, 1989, 1993a,b; Baraud, 2002).

**Theorem 3.1.** *Suppose the assumptions listed in Appendix A.2 hold, and let $\psi := p - q$. There exists a positive constant $C$ such that the condition*

$$\|\psi\|_2^2 \geq \|\psi - T_{h_{p,k}}\psi\|_2^2 + C \log\left(\frac{1}{\alpha}\right) \frac{\sqrt{C_k}}{\beta N}$$

*guarantees control over the probability of type II error, such that $\mathbb{P}_q\big(\Delta_\alpha^k(\mathbb{X}_N) = 0\big) \leq \beta$.*

Theorem 3.1, which is proved in Appendix I.1, provides a power guaranteeing condition consisting of two terms. The first term $\|\psi - T_{h_{p,k}}\psi\|_2^2$ indicates the size of the effect of the Stein integral transform operator on the difference in densities $\psi := p - q$, it is a measure of distance from the null (where this quantity is zero). The second term $C \log\big(1/\alpha\big)(\beta N)^{-1}\sqrt{C_k}$ is obtained from upper bounding the variance of the KSD $U$-statistic, it depends on the expectation of the squared Stein kernel $C_k := \mathbb{E}_{q,q}[h_{p,k}(X, Y)^2]$. This second term also controls the quantile of the test.

## 3.2 Aggregated test

We can now introduce our aggregated test, which is motivated by the earlier works of Fromont et al. (2012, 2013), Albert et al. (2022), and Schrab et al. (2021) for two-sample and independence testing.

We compute $\widetilde{K}_k^1, \ldots, \widetilde{K}_k^{B_2}$ further KSD values simulated from the null hypothesis obtained using either a parametric bootstrap or a wild bootstrap as in Equations (4) or (5), respectively. We consider a finite collection $\mathcal{K}$ of kernels satisfying the properties presented in Section 2. We construct an aggregated test $\Delta_\alpha^{\mathcal{K}}$, called KSDAGG, which rejects the null hypothesis if one of the single tests $\big(\Delta_{u_\alpha w_k}^k\big)_{k \in \mathcal{K}}$ rejects it, that is

$$\Delta_\alpha^{\mathcal{K}}(\mathbb{X}_N) := \mathbb{1}\big(\Delta_{u_\alpha w_k}^k(\mathbb{X}_N) = 1 \text{ for some } k \in \mathcal{K}\big).$$

The levels of the single tests are adjusted to ensure the aggregated test has the prescribed level $\alpha$. This adjustment is performed by introducing positive weights $(w_k)_{k \in \mathcal{K}}$ satisfying $\sum_{k \in \mathcal{K}} w_k \leq 1$ and some correction

$$u_\alpha := \sup\left\{ u \in \left(0, \min_{k \in \mathcal{K}} w_k^{-1}\right) : \widehat{P}_u \leq \alpha \right\} \tag{6}$$

---

**Algorithm 1** KSDAGG

---

**Inputs:** samples $\mathbb{X}_N = (x_i)_{i=1}^N$, density $p$ or score $\nabla \log p(\cdot)$, finite kernel collection $\mathcal{K}$, weights $(w_k)_{k \in \mathcal{K}}$, level $\alpha \in (0, e^{-1})$, estimation parameters $B_1, B_2, B_3 \in \mathbb{N}$, parametric or wild bootstrap

**Output:** 0 (fail to reject $\mathcal{H}_0$) or 1 (reject $\mathcal{H}_0$)

**Algorithm:**

**for** $k \in \mathcal{K}$ **do**

     compute $\bar{K}_k^{B_1+1} := \widehat{\mathrm{KSD}}_{p,k}^2(\mathbb{X}_N)$ as in Equation (1)

     compute $\left(\bar{K}_k^b\right)_{1 \le b \le B_1}$ as in Equations (4) or (5)

     sort $\left(\bar{K}_k^b\right)_{1 \le b \le B_1+1}$ in ascending order to obtain $\left(\bar{K}_k^{\bullet b}\right)_{1 \le b \le B_1+1}$

     compute $\left(\widetilde{K}_k^b\right)_{1 \le b \le B_2}$ as in Equations (4) or (5)

$u_{\min} = 0, \ u_{\max} = \min\limits_{k \in \mathcal{K}} w_k^{-1}$

**for** $t = 1, \dots, B_3$ **do**

$$u = \tfrac{1}{2}(u_{\min} + u_{\max}), \ \widehat{P}_u = \frac{1}{B_2} \sum_{b=1}^{B_2} \mathbb{1}\left( \max_{k \in \mathcal{K}} \left( \widetilde{K}_k^b - \bar{K}_k^{\bullet \lceil (B_1+1)(1-uw_k) \rceil} \right) > 0 \right)$$

     **if** $\widehat{P}_u \le \alpha$ **then** $u_{\min} = u$ **else** $u_{\max} = u$

$u_\alpha = u_{\min}$

**if** $\max\limits_{k \in \mathcal{K}} \left( \widehat{\mathrm{KSD}}_{p,k}^2(\mathbb{X}_N) - \bar{K}_k^{\bullet \lceil (B_1+1)(1-u_\alpha w_k) \rceil} \right) > 0$ **then return** 1 **else return** 0

**Time complexity:** $\mathcal{O}\big(|\mathcal{K}|(B_1 + B_2)N^2\big)$

---

where

$$\widehat{P}_u := \frac{1}{B_2} \sum_{b=1}^{B_2} \mathbb{1}\left( \max_{k \in \mathcal{K}} \left( \widetilde{K}_k^b - \bar{K}_k^{\bullet \lceil (B_1+1)(1-uw_k) \rceil} \right) > 0 \right)$$

is a Monte Carlo approximation of the type I error probability of the aggregated test with correction $u$

$$P_u := \mathbb{P}_p\left( \max_{k \in \mathcal{K}} \left( \widehat{\mathrm{KSD}}_{p,k}^2(\mathbb{X}_N) - \widehat{q}_{1-uw_k}^k \right) > 0 \right).$$

To compute $u_\alpha$, we estimate the supremum in Equation (6) by performing $B_3$ steps of the bisection method, the theoretical results account for this extra approximation. Detailed pseudocode for KSDAGG is provided in Algorithm 1, and details regarding our aggregation procedure are provided in Appendix G. We discuss potential limitations of KSDAgg in Appendix H.

We verify in the next proposition that performing this correction indeed ensures that our aggregated test $\Delta_\alpha^\mathcal{K}$ has the prescribed level $\alpha$.

**Proposition 3.2.** *For $\alpha \in (0, 1)$ and a collection of kernels $\mathcal{K}$, the aggregated test $\Delta_\alpha^\mathcal{K}$ satisfies*

$$\mathbb{P}_p\big(\Delta_\alpha^\mathcal{K}(\mathbb{X}_N) = 1\big) \le \alpha$$

*asymptotically using a wild bootstrap (with Lipschitz continuity of $h_{p,k}$ required) and non-asymptotically using a parametric bootstrap.*

The proof of Proposition 3.2 is presented in Appendix I.2. Details about the asymptotic result for the wild bootstrap case are reported in Appendix D. We now provide guarantees for the power of our aggregated test KSDAGG in terms of its uniform separation rate.

**Theorem 3.3.** *Suppose the assumptions listed in Appendix A.3 hold, and let $\psi := p - q$. There exists a positive constant $C$ such that if*

$$\|\psi\|_2^2 \ge \min_{k \in \mathcal{K}} \left( \|\psi - T_{h_{p,k}}\psi\|_2^2 + C \log\left(\frac{1}{\alpha w_k}\right) \frac{\sqrt{C_k}}{\beta N} \right)$$

*then the probability of type II error of $\Delta_\alpha^\mathcal{K}$ is controlled by $\beta$, that is, $\mathbb{P}_q\big(\Delta_\alpha^\mathcal{K}(\mathbb{X}_N) = 0\big) \le \beta$.*

The proof Theorem 3.3 can be found in Appendix I.3, it relies on upper bounding the probability of the intersection of some events by the minimum of the probabilities of each event, and on applying Theorem 3.1 after having verified that its assumptions are satisfied for the tests with adjusted levels. We observe that the aggregation procedure allows to achieve the smallest uniform separation rate of the single tests $\left(\Delta_\alpha^k\right)_{k \in \mathcal{K}}$ up to some logarithmic weighting term $\log(1/w_k)$.

### 3.3 Bandwidth selection

A specific application of the setting we have considered is the problem of bandwidth selection for a fixed kernel. Given a kernel $k : \mathbb{R}^d \times \mathbb{R}^d \to \mathbb{R}$, the function

$$k_\lambda(x, y) := k\left(\frac{x}{\lambda}, \frac{y}{\lambda}\right)$$

is also a kernel for any bandwidth $\lambda > 0$. A common example is the Gaussian kernel, for which we have $k(x, y) = \exp(-\|x - y\|_2^2)$ and $k_\lambda(x, y) = \exp\left(-\|x - y\|_2^2/\lambda^2\right)$. As shown by Gorham and Mackey (2017), a more appropriate kernel for goodness-of-fit testing using the KSD is the IMQ (inverse multiquadric) kernel, which is defined with $k(x, y) = \left(1 + \|x - y\|_2^2\right)^{-\beta_k}$ for a fixed parameter $\beta_k \in (0, 1)$ as

$$k_\lambda(x, y) = \left(1 + \frac{\|x - y\|_2^2}{\lambda^2}\right)^{-\beta_k} = \lambda^{2\beta_k}\left(\lambda^2 + \|x - y\|_2^2\right)^{-\beta_k} \propto \left(\lambda^2 + \|x - y\|_2^2\right)^{-\beta_k} \quad (7)$$

which is the well-known form of the IMQ kernel with parameters $\lambda > 0$ and $\beta_k \in (0, 1)$. Note that it is justified to consider the kernel up to a multiplicative constant because the single and aggregated tests are invariant under this kernel transformation.

In practice, as suggested by Gretton et al. (2012a), the bandwidth is often set to a heuristic such as the median or the standard deviation of the $L^2$-distances between the samples $(X_i)_{i=1}^N$, however, these are arbitrary choices with no theoretical guarantees. Another common approach proposed by Gretton et al. (2012b) for the linear-time setting, and extended to the quadratic-time setting by Liu et al. (2020), is to resort to data splitting in order to select a bandwidth on held-out data, by maximising for a proxy for asymptotic power (see Section 4.1 for details). Both methods were originally proposed for the two-sample problem, but extend straightforwardly to the goodness-of-fit setting.

By considering a kernel collection $\mathcal{K}_\Lambda = \{k_\lambda : \lambda \in \Lambda\}$ for a collection of bandwidths $\Lambda$, we can use our aggregated test KSDAGG to test multiple bandwidths using all the data and without resorting to arbitrary heuristics. We now obtain an expression for the uniform separation rate of $\Delta_\alpha^{\mathcal{K}_\Lambda}$ in terms of the bandwidths $\lambda \in \Lambda$.

**Corollary 3.4.** *Suppose the assumptions listed in Appendix A.3 hold for $\mathcal{K} = \mathcal{K}_\Lambda = \{k_\lambda : \lambda \in \Lambda\}$, and let $\psi := p - q$. There exists a positive constant $C$ such that the condition*

$$\|\psi\|_2^2 \geq \min_{\lambda \in \Lambda}\left(\left\|\psi - T_{h_{p,k_\lambda}}\psi\right\|_2^2 + C \log\left(\frac{1}{\alpha w_\lambda}\right)\frac{\sqrt{C_{k_\lambda}}}{\beta N}\right)$$

*ensures control over the probability of type II error of the aggregated test $\mathbb{P}_q\left(\Delta_\alpha^{\mathcal{K}_\Lambda}(\mathbb{X}_N) = 0\right) \leq \beta$.*

Corollary 3.4 follows from applying Theorem 3.3 to the collection of kernels $\mathcal{K}_\Lambda$. Our results hold with great generality as we have not imposed any restrictions on $\psi := p - q$ such as assuming it belongs to a specific regularity class. For this reason, the dependence on $\lambda$ in the terms $\|\psi - T_{h_{p,k_\lambda}}\psi\|_2^2$ and $\log\left(1/(\alpha w_\lambda)\right)(\beta N)^{-1}\sqrt{C_{k_\lambda}}$ is not explicit. In Section 3.4, we characterise this dependence with regularity assumptions on $\psi := p - q$, and derive uniform seperation rates in terms of the sample size.

### 3.4 Uniform separation rates over restricted Sobolev balls

In this section, we derive uniform separation rates with stronger assumptions on $p$ and $q$. This provides settings in which the power guaranteeing conditions of Theorems 3.1 and 3.3 are satisfied, and illustrates the interactions between the two terms in those conditions. We make the following assumptions.

- The model density $p$ is strictly positive on its connected and compact support $S \subseteq \mathbb{R}^d$.
- The score function $\nabla \log p(x)$ is continuous and bounded on $S$.
- The support of the density $q$ is a connected and compact subset of $S$.
- The kernel used is a scaled Gaussian kernel $k_\lambda(x, y) := \lambda^{2-d} \exp\left(-\|x - y\|_2^2/\lambda^2\right)$.

In particular, any strictly positive twice-differentiable density on $\mathbb{R}^d$ truncated to some $d$-dimensional interval will satisfy the assumptions for the model $p$. Any density truncated to the same $d$-dimensional interval will satisfy the assumption for the density $q$. As an example, one can consider truncated normal densities.

Given some smoothness parameter $s > 0$, radius $R > 0$ and dimension $d \in \mathbb{N} \setminus \{0\}$, the Sobolev ball $\mathcal{S}_d^s(R)$ is defined as the function space

$$\mathcal{S}_d^s(R) := \left\{ f \in L^1(\mathbb{R}^d) \cap L^2(\mathbb{R}^d) : \int_{\mathbb{R}^d} \|\xi\|_2^{2s} \left|\widehat{f}(\xi)\right|^2 d\xi \leq (2\pi)^d R^2 \right\},$$

where $\widehat{f}$ denotes the Fourier transform of $f$. For $s, t, d, R, L$ all strictly positive, we define the restricted Sobolev ball $\mathcal{S}_d^{s,t}(R, L)$ as containing all functions $f \in \mathcal{S}_d^s(R)$ satisfying

$$\int_{\|\xi\|_2 \leq t} \left|\widehat{f}(\xi)\right|^2 d\xi \leq \frac{1}{L} \int_{\mathbb{R}^d} \left|\widehat{f}(\xi)\right|^2 d\xi. \tag{8}$$

With the Sobolev assumption $p - q \in \mathcal{S}_d^s(R)$, the densities $p$ and $q$ can differ at any frequencies. The restricted Sobolev assumption $p - q \in \mathcal{S}_d^{s,t}(R, L)$ does not include the case in which the densities $p$ and $q$ differ only at low frequencies due to the additional restriction in Equation (8).

**Theorem 3.5.** *Suppose the assumptions listed in Appendix A.4 hold for $\mathcal{K}_\Lambda = \{k_\lambda : \lambda \in \Lambda\}$. Let $\psi := p - q$. We show that there exists some $L > 0$ such that if $p - q \in \mathcal{S}_d^{s,t}(R, L)$ then (i) & (ii) hold.*

*(i) Under the assumptions of Theorem 3.1, for the KSD test with bandwidth $\lambda := N^{-2/(4s+d)}$, the condition*

$$\|\psi\|_2^2 \geq C N^{-4s/(4s+d)}$$

*for some $C > 0$ guarantees control over the probability of type II error $\mathbb{P}_q\big(\Delta_\alpha^\lambda(\mathbb{X}_N) = 0\big) \leq \beta$.*

*(ii) Under the assumptions of Theorem 3.3, for KSDAGG with the collection*

$$\Lambda := \left\{ 2^{-\ell} : \ell \in \left\{1, \ldots, \left\lceil \frac{2}{d} \log_2\left( \frac{N}{\ln(\ln(N))} \right) \right\rceil \right\} \right\}$$

*and weights $w_\lambda := 6/\pi^2 \ell^2$, we have $\mathbb{P}_q\big(\Delta_\alpha^\Lambda(\mathbb{X}_N) = 0\big) \leq \beta$ provided that, for some $C > 0$,*

$$\|\psi\|_2^2 \geq C \left( \frac{N}{\ln(\ln(N))} \right)^{-4s/(4s+d)}.$$

The proof of Theorem 3.5 is presented in Appendix I.4. The uniform separation rate $N^{-4s/(4s+d)}$ in Theorem 3.5 is known to be optimal in the *minimax* sense over (unrestricted) Sobolev balls $\mathcal{S}_d^s(R)$ (Li and Yuan, 2019; Balasubramanian et al., 2021; Albert et al., 2022; Schrab et al., 2021). However, this rate is for the KSD test with bandwidth depending on the smoothness parameter $s > 0$ which is unknown in practice. Note that even methods which split the data to perform kernel selection are not able to select this bandwidth depending on $s$ and cannot achieve the minimax rate. The aggregation procedure is adaptive to this unknown parameter $s$: crucially, the collection of bandwidths $\Lambda$ does not depend on $s$, and so, the aggregated KSDAGG test of Theorem 3.5 (ii) can be implemented in practice, unlike the single KSD test of Theorem 3.5 (i). The price to pay for this adaptivity is only an iterated logarithmic factor in the minimax rate over Sobolev balls $\mathcal{S}_d^s(R)$. The uniform separation rates presented in Theorem 3.5 are over the restricted Sobolev balls $\mathcal{S}_d^{s,t}(R, L)$. Whether those rates can also be derived under less restrictive assumptions and in the more general setting of (unrestricted) Sobolev balls $\mathcal{S}_d^s(R)$, is a challenging problem, which is left for future work.

## 4 Implementation and experiments

We consider three different experiments based on a Gamma one-dimensional distribution, a Gaussian-Bernoulli Restricted Boltzmann Machine, and a Normalizing Flow for the MNIST dataset. We compare our proposed aggregated test KSDAGG against three alternatives: the KSD test which uses the median bandwidth, a test which splits the data to select an 'optimal' bandwidth according to a proxy for asymptotic test power, and a test which uses extra data for bandwidth selection. The 'extra data' test is designed simply to provide a best-case scenario for the bandwidth selection procedure which maximises asymptotic test power, but it cannot be used in practice as it has access to extra data compared to the other tests. In order to ensure that our tests always have correct levels for all bandwidth values, dimensions and sample sizes, we use the parametric bootstrap in our experiments.

### 4.1 Alternative bandwidth selection approaches

Gretton et al. (2012a) proposed to use the median heuristic as kernel bandwidth, it consists in the median of the $L^2$-distances between the samples given by

$$\lambda_{\mathrm{med}} \coloneqq \mathrm{median}\{\|x_i - x_j\|_2 : 0 \le i < j \le N\}.$$

Gretton et al. (2012b) first proposed, for the two-sample problem using a linear-time MMD estimator, to split the data and to use half of it to select an 'optimal' bandwidth which maximises a proxy for asymptotic power. This procedure was extended to quadratic-time estimators and to the goodness-of-fit framework by Jitkrittum et al. (2017), Sutherland et al. (2017) and Liu et al. (2020). These strategies rely on the asymptotic normality of the test statistic under $\mathcal{H}_a$. In our setting, the asymptotic power proxy to maximise is the ratio

$$\widehat{\mathrm{KSD}}_{p,k}^2(\mathbb{X}_N) \,/\, \widehat{\sigma}_{\mathcal{H}_a} \tag{9}$$

where $\widehat{\sigma}_{\mathcal{H}_a}^2$ is a closed-form regularised positive estimator of the asymptotic variance of $\widehat{\mathrm{KSD}}_{p,k}^2$ under $\mathcal{H}_a$ (Liu et al., 2020, Equation 5). In our experiments, we also consider a test which has access to $N$ extra samples drawn from $q$ to select an 'optimal' bandwidth to run the KSD test on the original $N$ samples $\mathbb{X}_N$. This test is interesting to compare to because it uses an 'optimal' bandwidth without being detrimental to power (as it uses all $N$ samples $\mathbb{X}_N$ to run the test with the selected bandwidth).

### 4.2 Experimental details

Inspired from Theorem 3.5, in each experiments, we use KSDAGG with different collections of bandwidths of the form $\Lambda(\ell_-, \ell_+) \coloneqq \{2^i \lambda_{\mathrm{med}} : i = \ell_-, \ldots, \ell_+\}$ for the median bandwidth $\lambda_{\mathrm{med}}$ and integers $\ell_- < \ell_+$ with uniform weights $w_\lambda \coloneqq 1/(\ell_+ - \ell_- + 1)$. For the tests which split the data, we select the bandwidth, out of the collection $\Lambda(\ell_-, \ell_+)$, which maximises the power proxy discussed in Section 4.1. All our experiments are run with level $\alpha = 0.05$ using the IMQ kernel defined in Equation (7) with parameter $\beta_k = 0.5$. We use a parametric bootstrap with $B_1 = B_2 = 500$ bootstrapped KSD values to compute the adjusted test thresholds, and $B_3 = 50$ steps of bisection method to estimate the correction $u_\alpha$ in Equation (6). For our aggregated test, we have also designed another collection which is parameter-free (no varying parameters across experiments) and performs as well as the previous variant whose parameters are chosen appropriately for each experiments. The collection is obtained using the maximal inter-sample distance normalised by the dimension (see details in Appendix B). We denote the resulting robust test by KSDAGG$^\star$ for which we use either a wild or parametric bootstrap with $B_1 = B_2 = 2000$ and $B_3 = 50$, this is the variant we recommend using in practice. To estimate the probability of rejecting the null hypothesis, we average the test outputs across 200 repetitions. All experiments have been run on an AMD Ryzen Threadripper 3960X 24 Cores 128Gb RAM CPU at 3.8GHz, the runtime is of the order of a couple of hours (significant speedup can be obtained by using parallel computing). We have used the implementation of Jitkrittum et al. (2017) to sample from a Gaussian-Bernoulli Restricted Boltzmann Machine, and Phillip Lippe's implementation of MNIST Normalizing Flows, both under the MIT license.

### 4.3 Gamma distribution

For our first experiment, we consider a one-dimensional Gamma distribution with shape parameter 5 and scale parameter 5 as the model $p$. For $q$, we draw 500 samples from a Gamma distribution with the same scale parameter 5 and with a shifted shape parameter $5 + s$ for $s \in \{0, 0.1, 0.2, 0.3, 0.4\}$. We consider the collection of bandwidths $\Lambda(0, 10)$.

The results we obtained are presented in Figure 1a. We observe that all tests have the prescribed level 0.05 under the null hypothesis, which corresponds to the case $s = 0$. As the shift parameter $s$ increases, the two densities $p$ and $q$ become more different and rejection of the null becomes an easier task, thus the test power increases. Our aggregated test KSDAGG achieves the same power as the KSD test with an 'optimal' bandwidth selected using extra data by maximizing the proxy for asymptotic power. The median test obtains only slightly lower power, this closeness in power can be explained by the fact that this one-dimensional problem is a simple one. All four tests KSDAGG $\Lambda(0, 10)$, KSDAGG$^\star$, KSD median, and KSD split, all achieve very similar power. We note that the normal splitting test has significantly lower power: this is because, even though it uses an 'optimal' bandwidth, it is then run on only half the data, which results in a loss of power.

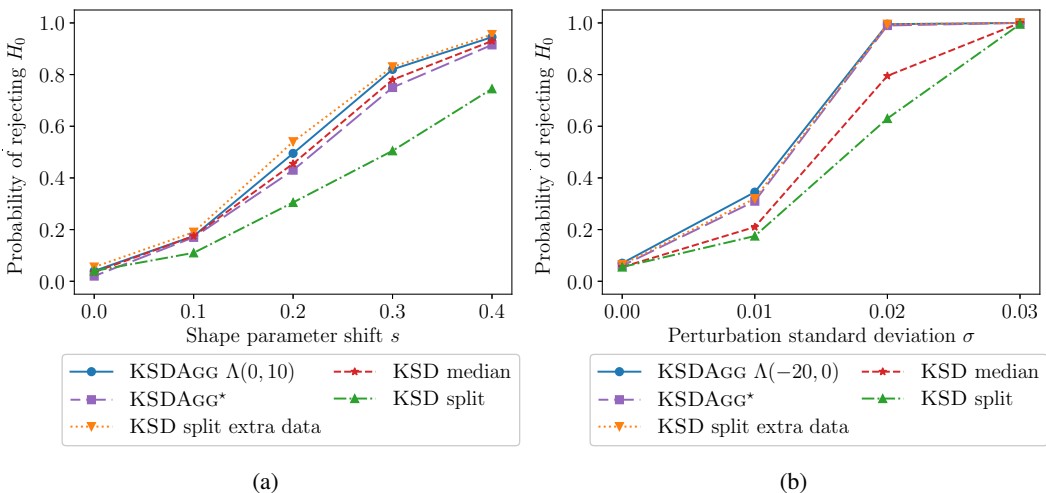

Figure 1: (a) Gamma distribution. (b) Gaussian-Bernoulli Restricted Boltzmann Machine.

## 4.4 Gaussian-Bernoulli Restricted Boltzmann Machine

As first considered by Liu et al. (2016) for goodness-of-fit testing using the KSD, we consider a Gaussian-Bernoulli Restricted Boltzmann Machine. It is a graphical model with a binary hidden variable $h \in \{-1, 1\}^{d_h}$ and a continuous observable variable $x \in \mathbb{R}^d$. Those variables have joint density

$$p(x, h) \;=\; \frac{1}{Z} \exp\left(\frac{1}{2} x^\top B h + b^\top x + c^\top h - \frac{1}{2}\|x\|_2^2\right)$$

where $Z$ is an unknown normalizing constant. By marginalising over $h$, we obtain the density $p$ of $x$

$$p(x) \;=\; \sum_{h \in \{-1,1\}^{d_h}} p(x, h).$$

We can sample from it using a Gibbs sampler with 2000 burn-in iterations. We use the dimensions $d = 50$ and $d_h = 40$ as considered by Jitkrittum et al. (2017) and Grathwohl et al. (2020). Even though computing $p$ is intractable for large dimension $d_h$, the score function admits a convenient closed-form expression

$$\nabla \log p(x) \;=\; b - x + B \frac{\exp\big(2(B^\top x + c)\big) - 1}{\exp(2(B^\top x + c)) + 1}.$$

We draw the components of $b$ and $c$ from Gaussian standard distributions and sample Rademacher variables taking values in $\{-1, 1\}$ for the elements of $B$ for the model $p$. We draw 1000 samples from a distribution $q$ which is constructed in a similar way as $p$ but with the difference that some Gaussian noise $\mathcal{N}(0, \sigma)$ is injected into each of the elements of $B$. For the standard deviations of the perturbations, we consider $\sigma \in \{0, 0.01, 0.02, 0.03\}$. We use the collection $\Lambda(-20, 0)$ (different from Section 4.3), for KSDAGG$^\star$ we use a wild bootstrap, the results are provided in Figure 1b.

Again, we observe that our aggregated tests KSDAGG $\Lambda(0, 20)$ and KSDAGG$^\star$ match the power obtained by the test which uses extra data to select an 'optimal' bandwidth. This means that, in this experiment, the aggregated tests obtain the same power as the 'best' single test. The difference with the median heuristic test is significant in this experiment, and the splitting test obtains the lowest power of the four tests. Again, all tests have well-calibrated levels (case $\sigma = 0$) and increasing the noise level $\sigma$ results in more power for all the tests.

## 4.5 MNIST Normalizing Flow

Finally, we consider a high-dimensional problem working with images in dimensions $28^2 = 784$. We consider a multi-scale Normalizing Flow (Dinh et al., 2017; Kingma and Dhariwal, 2018) which has been trained on the MNIST dataset (LeCun et al., 1998, 2010), it is a generative model which has a

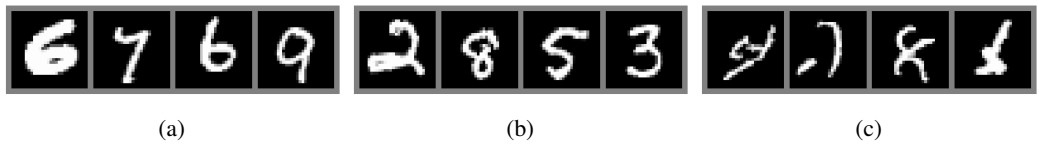

(a)                                        (b)                                        (c)

Figure 2: (a) Digits from the MNIST dataset. (b, c) Digits sampled from the Normalizing Flow.

probability density $p$. As observed in Figure 2, some samples produced by the model can look exactly like MNIST digits, while others do not resemble digits. This Normalizing Flow has been trained to 'ideally' produce samples from the MNIST dataset. We are interested in whether or not we can detect the difference in densities. Given some images of digits, are we able to tell if those were sampled from the Normalizing Flow model? We consider the case where the samples from $q$ are drawn from the true MNIST dataset (power experiment), and the case where the images from $q$ are sampled from the Normalizing Flow model (level experiment, confirming performance for the power experiment). The experiments are run with the collection of bandwidths $\Lambda(-20, 0)$. The results are displayed in Figure 3a and Figure 3b. We use a parametric bootstrap for KSDAGG$^\star$.

In Figure 3a, we observe that the four tests have correct levels (around 0.05) for the five different sample sizes considered (the small fluctuations about the designed test level can be explained by the fact that we are averaging 200 test outputs to estimate these levels). The well-calibrated levels obtained in Figure 3a demonstrate the validity of the power results presented in Figure 3b.

As seen in Figure 3b, only our aggregated tests KSDAGG and KSDAGG$^\star$ obtain high power; they are able to detect that MNIST samples are not drawn from the Normalizing Flow. The power of the other tests increases only marginally as the sample size increases. The test which uses extra data to select an 'optimal' bandwidth performs poorly when compared to the aggregated tests. This could be explained by the fact that this test selects the bandwidth using a proxy for the asymptotic power, and that in this high-dimensional setting, the asymptotic regime is not attained with sample sizes below 500. See Appendices E and F for details regarding selected bandwidths and for reported runtimes.

| Sample size | KSDAGG $\Lambda(-20, 0)$ | KSDAGG$^\star$ | KSD median | KSD split | KSD split extra data |
|---|---|---|---|---|---|
| 100 | 0.05 | 0.035 | 0.04 | 0.05 | 0.04 |
| 200 | 0.06 | 0.03 | 0.055 | 0.05 | 0.055 |
| 300 | 0.045 | 0.055 | 0.085 | 0.05 | 0.085 |
| 400 | 0.04 | 0.03 | 0.055 | 0.03 | 0.055 |
| 500 | 0.035 | 0.035 | 0.045 | 0.04 | 0.045 |

(a)

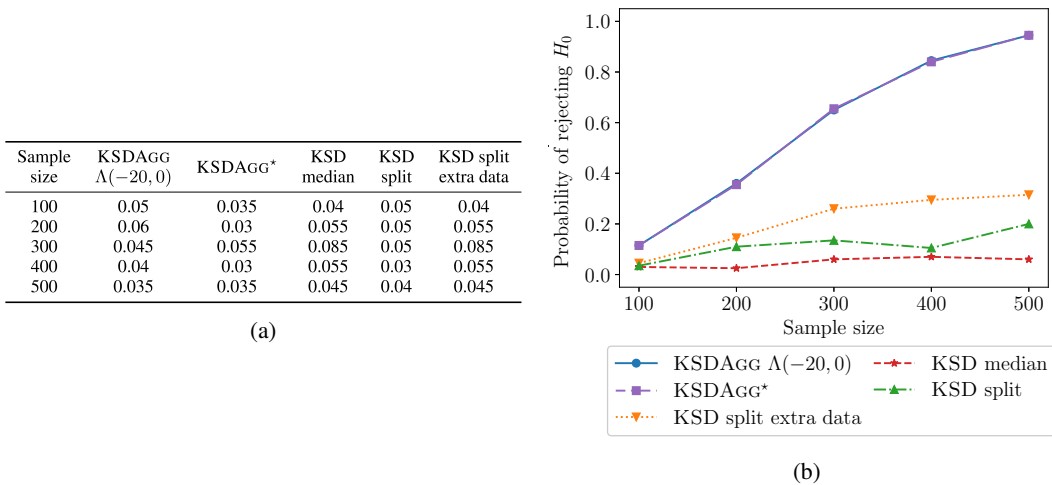

(b)

Figure 3: Normalizing Flow MNIST. (a) Level experiment. (b) Power experiment.

# 5   Acknowledgements

Antonin Schrab acknowledges support from the U.K. Research and Innovation (EP/S021566/1). Benjamin Guedj acknowledges partial support by the U.S. Army Research Laboratory and the U.S. Army Research Office, and by the U.K. Ministry of Defence and the U.K. Engineering and Physical Sciences Research Council (EP/R013616/1), and by the French National Agency for Research (ANR-18-CE40-0016-01 & ANR-18-CE23-0015-02). Arthur Gretton acknowledges support from the Gatsby Charitable Foundation.

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
