# Supplementary material for 'KSD Aggregated Goodness-of-fit Test'

## A   Assumptions

### A.1   Assumptions common to all theoretical results

- kernel $k$ is $C_0$-universal (Carmeli et al., 2010, Definition 4.1)
- $\mathrm{KSD}^2_{p,k}(q) = \mathbb{E}_{q,q}[h_{p,k}(X, Y)] < \infty$
- $C_k \geq 1$, where $C_k \coloneqq \mathbb{E}_{q,q}[h_{p,k}(X, Y)^2] < \infty$ as defined in Equation (2)
- $\|q\|_\infty \leq M$ for some $M > 0$
- $\mathbb{E}_q \Big[ \Big\| \nabla \Big( \log \frac{p(X)}{q(X)} \Big) \Big\|_2^2 \Big] < \infty$
- $\alpha \in (0, e^{-1})$
- $\beta \in (0, 1)$

### A.2   Assumptions for Theorem 3.1

- Assumptions of Appendix A.1
- $B_1 \geq 3\big(\log(8/\beta) + \alpha(1 - \alpha)\big)/\alpha^2$
- assumption for parametric bootstrap: $n/\sqrt{C_k} \geq \ln(1/\alpha)$
- assumption for wild bootstrap: Lipschitz continuity of $h_{p,k}$
- constant $C$ depends on $M$ and $d$

### A.3   Assumptions for Theorem 3.3 and Corollary 3.4

- Assumptions of Appendix A.1
- $B_1 \geq 12\Big(\max_{k \in \mathcal{K}} w_k^{-2}\Big)\big(\log(8/\beta) + \alpha(1 - \alpha)\big)/\alpha^2$
- $B_2 \geq 8\log\big(2/\beta\big)/\alpha^2$
- $B_3 \geq \log_2\big(4\min_{k \in \mathcal{K}} w_k^{-1}/\alpha\big)$
- positive weights $(w_k)_{k \in \mathcal{K}}$ satisfying $\sum_{k \in \mathcal{K}} w_k \leq 1$
- assumption for parametric bootstrap: $n/\sqrt{C_k} \geq \ln(1/\alpha w_k)$ for $k \in \mathcal{K}$
- assumption for wild bootstrap: Lipschitz continuity of $h_{p,k}$
- constant $C$ depends on $M$ and $d$

### A.4   Assumptions for Theorem 3.5

- Assumptions of Appendix A.1
- $B_1 \geq 12\Big(\max_{\lambda \in \Lambda} w_{k_\lambda}^{-2}\Big)\big(\log(8/\beta) + \alpha(1 - \alpha)\big)/\alpha^2$
- $B_2 \geq 8\log\big(2/\beta\big)/\alpha^2$
- $B_3 \geq \log_2\big(4\min_{\lambda \in \Lambda} w_{k_\lambda}^{-1}/\alpha\big)$
- assumption for parametric bootstrap: $n/\sqrt{C_{k_\lambda}} \geq \ln(1/\alpha w_{k_\lambda})$ for $\lambda \in \Lambda$
- assumption for wild bootstrap: Lipschitz continuity of $h_{p,k_\lambda}$
- constant $C$ is independent of the sample size $N$

## B    Collection of bandwidths for KSDAGG$^\star$

In this section, we explain how the collection for the robust test KSDAGG$^\star$ is constructed. First, we compute the maximal inter-sample distance

$$\lambda_{\mathsf{M}} := \max\big\{\|x_i - x_j\|_2 : 0 \le i < j \le N\big\},$$
$$\lambda_{\max} := \max\big\{\lambda_{\mathsf{M}}, 2\big\}.$$

The collection of $B$ bandwidths is then defined as

$$\Big\{d^{-1}\lambda_{\max}^{(i-1)/(B-1)} : i = 1, \dots, B\Big\}$$

where $d$ is the dimension of the samples. This collection is a discretisation of the interval $\big[d^{-1}, d^{-1}\lambda_{\max}\big]$. In the limit as the number of bandwidths $B$ goes to infinity, the collection becomes the continuous interval. In practice, we use $B = 10$ bandwidths and we observed that increasing $B$ in all three experiments ($B = 100, 200, \dots$) does not affect the power, that is, by using 10 bandwidths we obtain the same power as if we were to consider the aggregated test with the continuous collection $\big[d^{-1}, d^{-1}\lambda_{\max}\big]$ of bandwidths.

## C    Background on Kernel Stein Discrepancy

**Background.** Stein's methods (Stein, 1972) have been widely used in the machine learning and statistics communities. At the heart of this field, for the Langevin Stein operator $\mathcal{A}_p$ defined as

$$(\mathcal{A}_p f)(x) := f(x)^\top \nabla \log p(x) + \mathrm{Tr}\big(\nabla f(x)\big),$$

is the fact that smooth densities $p$ and $q$ are equal if and only if

$$\mathbb{E}_q\big[(\mathcal{A}_p f)(x)\big] = 0$$

for smooth functions $f$ vanishing at the boundaries. This is known as Stein's identity (Stein, 1972; Stein et al., 2004), and also holds more generally for other Stein operators. Gorham and Mackey (2015) utilised this identity to define Stein discrepancies as

$$\sup_{f \in \mathcal{F}} \mathbb{E}_q\big[f(x)^\top \nabla \log p(x) + \mathrm{Tr}\big(\nabla f(x)\big)\big]$$

for some space of smooth functions $\mathcal{F}$ satisfying Stein's identity. As proposed by Oates and Girolami (2016), choosing $\mathcal{F}$ to be the unit ball in a Reproducing Kernel Hilbert Space (Aronszajn, 1950, RKHS) denoted $\mathcal{H}$, that is $\mathcal{F} := \{f : \|f\|_{\mathcal{H}} \le 1\}$, we obtain the Kernel Stein Discrepancy (KSD; Liu et al., 2016; Chwialkowski et al., 2016). The KSD can also be expressed in terms of the Stein kernel as presented in Section 2.

**Goodness-of-fit related work.** Stein methods have been used to construct goodness-of-fit tests in various types of data: directional data (Xu and Matsuda, 2020), time-to-event data (Fernandez et al., 2020), data on Riemannian manifolds (Xu and Matsuda, 2021), conditional data (Jitkrittum et al., 2020) , graph data (Weckbecker et al., 2022; Xu and Reinert, 2021, 2022a), functional data (Wynne et al., 2022; Wynne and Duncan, 2022), and generative data (Xu and Reinert, 2022b). We also point out the works of Fernandez and Gretton (2019) on an MMD-based goodness-of-fit test for censored data, of Key et al. (2021) on composite goodness-of-fit tests using either the MMD or KSD, of Jitkrittum et al. (2017) on a linear-time KSD goodness-of-fit test, of Gorham and Mackey (2017) on measuring sample quality with kernels and on KSD dominating weak convergence for some kernels, of Huggins and Mackey (2018) on random feature Stein discrepancies, of Korba et al. (2021) on the KSD Wasserstein gradient flow, of Oates et al. (2019) on convergence rates for a class of estimators based on Stein's method, and of Barp et al. (2019) on minimum Stein discrepancy estimators. Lim et al. (2019) and Kanagawa et al. (2019) use KSD tests for multiple model comparison and for comparing latent variable models, respectively. It is worth noting those score-based methods suffer from blindness to isolated components and mixing proportions (Wenliang and Kanagawa, 2020; Zhang et al., 2022). Xu (2021, 2022) present unified approaches for goodness-of-fit testing, and Fernandez and Rivera (2022) introduce a general framework for analysing kernel-based tests working directly with random functionals on Hilbert spaces. Finally, Anastasiou et al. (2021) provide a review of recent developments based on Stein's methods.

## D  Asymptotic level for KSD test with wild bootstrap

The proof of asymptotic level of the KSD test using a wild bootstrap was proved by Chwialkowski et al. (2016, Proposition 3.2) relying on the result of Leucht (2012, Theorem 2.1) and of and Chwialkowski et al. (2014, Lemma 5). Their results for $V$-statistics can be extended to the use of $U$-statistics. They proved this result for the more general non-i.i.d. case, while we need only the result for the simpler i.i.d. case. They proved that the difference between true quantiles and the wild bootstrap quantiles converges to zero in probability under the null hypothesis with the following dependence on $N$:

$$\sup_{t \in \mathbb{R}} \left| \mathbb{P}(NB_N > t \mid \mathbb{X}_N) - \mathbb{P}(NK_N > t) \right|$$

converges to $0$ in probability, where $K_N$ is the KSD estimator of Equation (1), and $B_N$ is the wild bootstrap KSD of Equation (5). Lipschitz continuity of the Stein kernel $h_{p,k}$ (defined in Section 2) is required for the result to hold.

## E  Runtimes comparison of all KSD tests considered

The time complexity of KSDAGG is $\mathcal{O}\big(|\mathcal{K}|(B_1 + B_2)N^2\big)$ as provided in Algorithm 1. It grows linearly with the number of kernels $|\mathcal{K}|$, quadratically with the sample size $N$, and linearly with the number of bootstrap samples $B_1 + B_2$.

Table 1: Runtimes (averaged over 10 runs and reported in seconds) for the Normalizing Flow MNIST experiment presented in Section 4.5 using a wild bootstrap.

| SAMPLE SIZE | KSDAGG | KSD MEDIAN | KSD SPLIT | KSD SPLIT EXTRA DATA |
|---|---|---|---|---|
| 100 | 0.037 | 0.005 | 0.022 | 0.023 |
| 200 | 0.084 | 0.010 | 0.064 | 0.070 |
| 300 | 0.162 | 0.020 | 0.132 | 0.145 |
| 400 | 0.276 | 0.034 | 0.230 | 0.253 |
| 500 | 0.421 | 0.051 | 0.359 | 0.395 |

We report in Table 1 the runtimes of KSDAGG, KSD median, KSD split, and KSD split extra data, all using a wild bootstrap to estimate the quantiles in the MNIST Normalizing Flow experimental setting considered in Section 4.5.

Since we consider a collection of bandwidths $\Lambda$, the time complexity of KSDAGG is $\mathcal{O}(|\Lambda|(B_1 + B_2)N^2)$ and the one for median KSD is $\mathcal{O}(B_1 N^2)$ where, in the setting considered, we have $|\Lambda| = 21$ and $B_1 = B_2 = 500$. While KSDAGG takes roughly 10 times longer to run than KSD median, we could have expected a larger difference looking at the time complexities. This can be explained by the fact that there are two major time-consuming steps: (i) computing the kernel matrices, and (ii) computing the wild bootstrap samples. While (i) has complexity $\mathcal{O}(N^2)$ and (ii) complexity $\mathcal{O}(BN^2 + NB^2)$, the constant for step (i) is much larger than the one for step (ii) since (i) requires computing the matrix of pairwise distances while (ii) only consists in computing some matrix multiplications. Note that to compute the $|\Lambda|$ kernel matrices for KSDAGG, we need to compute the matrix of pairwise distances only once.

When splitting the data, the computationally expensive step is to select the bandwidth. All the $|\Lambda|$ kernel matrices need to be computed, as for KSDAGG, which is the expensive step (i). The KSD split test runs only slightly faster than the KSD split extra test; it runs faster than KSDAGG but their runtimes are of the same order of magnitude.

## F  Details on MNIST Normalizing Flow experiment

We provide details about the results reported in Figure 3b of the MNIST Normalizing Flow experiment presented in Section 4.5. In that case, the median bandwidth $\lambda_{\mathrm{med}}$ is on average 2437. The collection of bandwidths considered for KSDAGG is $\Lambda(-20, 0) \coloneqq \big\{2^i \lambda_{\mathrm{med}} : i = -20, \ldots, 0\big\}$. When

KSDAGG rejects the null hypothesis, the smallest bandwidth $2^{-20}\lambda_{\mathrm{med}} \approx 2^{-20} \cdot 2437 \approx 0.002$ (among others) rejects the single test with adjusted level. The bandwidth selected by KSD split extra data is the largest bandwidth $2^0\lambda_{\mathrm{med}} \approx 2437$. The bandwidth selection is performed by maximizing the criterion in Equation (9). Sutherland et al. (2017) showed that this is equivalent to maximizing asymptotic power in the case of the MMD; the same result holds straightforwardly for the KSD due to similar asymptotic properties. The criterion only maximizes asymptotic power and has no guarantee when using limited data. In this high-dimensional setting ($d = 784$) with sample sizes smaller than $500$, the asymptotic regime is not attained, and the criterion used for bandwidth selection does not maximize the power in this non-asymptotic setting. So, even though split extra has access to some extra data, it does not have an accurate criterion to select the bandwidth and ends up selecting the largest bandwidth, which is not well-adapted to this problem. This explains the low power obtained by KSD split extra data.

## G  Details on our aggregation procedure

**More powerful than Bonferroni correction.** The Bonferroni correction for multiple testing corresponds to using the adjusted level $\alpha/|\mathcal{K}|$ for each of the $|\mathcal{K}|$ tests. With the correction used for KSDAGG with uniform weights, the adjusted level for each test is $u_\alpha/|\mathcal{K}|$ with $u_\alpha$ defined in Equation (6). It can be shown that $u_\alpha \geq \alpha$ (Albert et al., 2022, Lemma 4), which means that KSDAGG will always reject the null when the test with Bonferroni correction would reject it. When using uniform weights, the multiple testing correction we use is guaranteed to always lead to a test as least as powerful as the one using the Bonferroni correction. The proof that the Bonferroni correction guarantees control of the probability of type I error uses a loose union bound argument, the method we use aims to tighten this loose upper bound. We present edge-case examples to illustrate the strengths of our multiple testing correction. First, assume that the $|\mathcal{K}|$ events are all disjoints, then the union bound is tight and both methods yield the adjusted levels $\alpha/|\mathcal{K}|$. Second, assume that all events are the same (*e.g.* same kernels), then the Bonferroni correction still yields adjusted levels $\alpha/|\mathcal{K}|$, while our multiple testing strategy can detect that there is nothing to correct for and provide 'adjusted' levels $\alpha$. We note that, given some fixed weights and level, there does not exist a single $u_\alpha$ associated to them, it will depend on what the events are.

**Choice of weights.** Without any prior knowledge (which is often the case in practice), we recommend using uniform weights since we do not expect any particular kernels/bandwidths to be better-suited than others. If the user has some prior knowledge of which kernels/bandwidths would be better for the task considered, then those should be given higher weights. We allow for weights whose sum is strictly smaller than 1 simply for the convenience of being able to add a new kernel/bandwidth with a new weight without changing the previous weights (as in Theorem 3.5 with weights $6/\pi^2\ell^2$ for $\ell \in \mathbb{N} \setminus \{0\}$). Multiplying all the weights by a constant simply results in dividing the correction $u_\alpha$ defined in Equation (6) by the same constant. This means that the product $u_\alpha w_\lambda$ remains the same, and hence the definition of the aggregated test is not affected by this change. For simplicity, in practice, we use weights whose sum is equal to 1.

**Interpretability.** When KSDAGG rejects the null hypothesis, we can check which specific kernels reject the adjusted tests: this provides the user with information about the kernels which are well-adapted to the problem considered. The 'best' selection of kernels is naturally returned as a side-effect of the test (without requiring data splitting). This contributes to the interpretability of KSDAGG, for instance in the case where different kernels prioritise different features.

**Extension to uncountable collections of kernels.** Our KSDAGG test aggregates over a finite collection of kernels. While even for large collections of kernels KSDAGG retains high power (due our multiple testing correction), it would be theoretically interesting to be able to consider an uncountable/continuous collection of kernels (parametrised by the kernel bandwidth on the positive real line for example). Optimizing the kernel parameters continuously can be done by selecting the parameters which maximise a proxy for test power as in Equation (9), but to the best of our knowledge, this has currently never been done without data splitting (which usually result in a loss of power). The extension to the case of uncountable collections remains a very challenging problem, and it would be of great theoretical interest to derive such a method in the future. However, as shown empirically in Appendix B, with a discretisation using only 10 bandwidths, our aggregated test KSDAGG$^\star$ already obtains the same power as the test using the continuous interval (in limit of the discretisation).

**Extension to KSD tests with adaptive features.** Our proposed test, KSDAGG, provides a solution to the problem of KSD adaptivity in the goodness-of-fit framework without requiring data splitting. A potential future direction of interest could be to tackle the adaptivity problem of the KSD-based linear-time goodness-of-fit test proposed by Jitkrittum et al. (2017). In this setting, the data is split to select feature locations (and the kernel bandwidth), the KSD test is then run using those adaptive features. A challenging problem would be to obtain those adaptive features using an aggregation procedure which avoids splitting the data.

## H  Adversarially constructed settings in which KSDAGG **might fail**

**Median bandwidth test.** In settings (possibly adversarially constructed) where the median bandwidth is the 'best' bandwidth, the KSD test with median bandwidth could in theory be more competitive than KSDAGG since by considering a large collection of bandwidths we are not only considering the 'best' median bandwidth, but also 'worse' bandwidths. However, in practice, we cannot know in advance which bandwidth would perform well, and KSDAGG retains power even for large collections of bandwidths (21 bandwidths considered in MNIST Normalizing Flow experiment in Section 4.5).

**Data splitting test.** Another setting (also possibly adversarially constructed) could be one in which the 'best' bandwidth lies in between two bandwidths of our collection which are 'worse' bandwidths. In that case, a test which uses data splitting to select an 'optimal' bandwidth would be able to select it, however, one must bear in mind the loss of power due to data splitting. In such a setting, the power of KSDAGG could then be improved by considering a finer collection of bandwidths. However, this issue did not arise in our experiments where we found that the aggregation procedure outperformed the competing approaches.

## I  Proofs

### I.1  Proof of Theorem 3.1

Note that

$$
\begin{aligned}
\mathbb{P}_q\big(\Delta_\alpha^k(\mathbb{X}_N) \,=\, 0\big) \,&=\, \mathbb{P}_q\Big(\widehat{\mathrm{KSD}}_{p,k}^2(\mathbb{X}_N) \,\leq\, \widehat{q}_{1-\alpha}^k\Big) \\
&=\, \mathbb{P}_q\Big(\widehat{\mathrm{KSD}}_{p,k}^2(\mathbb{X}_N) - \mathrm{KSD}_{p,k}^2(q) \,\leq\, \widehat{q}_{1-\alpha}^k - \mathrm{KSD}_{p,k}^2(q)\Big)
\end{aligned}
$$

where

$$
\mathrm{KSD}_{p,k}^2(q) \,=\, \mathbb{E}_q\Big[\widehat{\mathrm{KSD}}_{p,k}^2(\mathbb{X}_N)\Big]
$$

since $\widehat{\mathrm{KSD}}_{p,k}^2$ is an unbiased estimator. By Chebyshev's inequality (Chebyshev, 1899), we know that

$$
\begin{aligned}
\beta \,\geq\, &\mathbb{P}_q\left(\left|\mathrm{KSD}_{p,k}^2(q) - \widehat{\mathrm{KSD}}_{p,k}^2(\mathbb{X}_N)\right| \,\geq\, \sqrt{\frac{\mathrm{var}_q\Big(\widehat{\mathrm{KSD}}_{p,k}^2(\mathbb{X}_N)\Big)}{\beta}}\right) \\
\,\geq\, &\mathbb{P}_q\left(\mathrm{KSD}_{p,k}^2(q) - \widehat{\mathrm{KSD}}_{p,k}^2(\mathbb{X}_N) \,\geq\, \sqrt{\frac{\mathrm{var}_q\Big(\widehat{\mathrm{KSD}}_{p,k}^2(\mathbb{X}_N)\Big)}{\beta}}\right) \\
\,=\, &\mathbb{P}_q\left(\widehat{\mathrm{KSD}}_{p,k}^2(\mathbb{X}_N) - \mathrm{KSD}_{p,k}^2(q) \,\leq\, -\sqrt{\frac{\mathrm{var}_q\Big(\widehat{\mathrm{KSD}}_{p,k}^2(\mathbb{X}_N)\Big)}{\beta}}\right).
\end{aligned}
$$

We deduce that $\mathbb{P}_q\big(\Delta_\alpha^k(\mathbb{X}_N) = 0\big) \le \beta$ if

$$\widehat{q}_{1-\alpha}^k - \mathrm{KSD}_{p,k}^2(q) \le -\sqrt{\frac{\mathrm{var}_q\Big(\widehat{\mathrm{KSD}}_{p,k}^2(\mathbb{X}_N)\Big)}{\beta}}$$

$$\mathrm{KSD}_{p,k}^2(q) \ge \widehat{q}_{1-\alpha}^k + \sqrt{\frac{\mathrm{var}_q\Big(\widehat{\mathrm{KSD}}_{p,k}^2(\mathbb{X}_N)\Big)}{\beta}}. \tag{10}$$

The condition in Equation (10), which controls the probability of type I error of $\Delta_\alpha^k$, needs to hold with high probability since $\widehat{q}_{1-\alpha}^k$ depends on the randomness of, either the new samples drawn from $p$ for the parametric bootstrap in Equation (4), or of the Rademacher random variables for the wild bootstrap in Equation (5).

We want to derive a condition in terms of $\|p - q\|_2^2$ rather than in terms of $\mathrm{KSD}_{p,k}^2(q)$ as in Equation (10). For this, using Stein's identity, we obtain

$$\begin{aligned}
\langle \psi, T_{h_{p,k}}\psi\rangle_2 &= \int_{\mathbb{R}^d} \psi(y)\big(T_{h_{p,k}}\psi\big)(y)\,\mathrm{d}y \\
&= \int_{\mathbb{R}^d}\int_{\mathbb{R}^d} h_{p,k}(x,y)\psi(x)\psi(y)\,\mathrm{d}x\,\mathrm{d}y \\
&= \int_{\mathbb{R}^d}\int_{\mathbb{R}^d} h_{p,k}(x,y)(p(x) - q(x))(p(y) - q(y))\,\mathrm{d}x\,\mathrm{d}y \\
&= \int_{\mathbb{R}^d}\int_{\mathbb{R}^d} h_{p,k}(x,y)q(x)q(y)\,\mathrm{d}x\,\mathrm{d}y \\
&= \mathbb{E}_{q,q}[h_{p,k}(X,Y)] \\
&= \mathrm{KSD}_{p,k}^2(q)
\end{aligned}$$

which gives

$$\mathrm{KSD}_{p,k}^2(q) = \langle \psi, T_{h_{p,k}}\psi\rangle = \frac{1}{2}\Big(\|\psi\|_2^2 + \big\|T_{h_{p,k}}\psi\big\|_2^2 - \big\|\psi - T_{h_{p,k}}\psi\big\|_2^2\Big).$$

To guarantee $\mathbb{P}_q\big(\Delta_\alpha^k(\mathbb{X}_N) = 0\big) \le \beta$, an equivalent condition to the one presented in Equation (10) is then

$$\|\psi\|_2^2 \ge \big\|\psi - T_{h_{p,k}}\psi\big\|_2^2 - \big\|T_{h_{p,k}}\psi\big\|_2^2 + 2\widehat{q}_{1-\alpha}^k + 2\sqrt{\frac{\mathrm{var}_q\Big(\widehat{\mathrm{KSD}}_{p,k}^2(\mathbb{X}_N)\Big)}{\beta}} \tag{11}$$

which needs to hold with high probability over the randomness in $\widehat{q}_{1-\alpha}^k$. We now upper bound the quantile and variance terms in Equation (11) to obtain a new condition ensuring control of the type II error.

To bound the quantile term $\widehat{q}_{1-\alpha}^k$ using $B_1$ wild bootstrapped statistics, using the Dvoretzky–Kiefer–Wolfowitz inequality (Dvoretzky et al., 1956; Massart, 1990) as done by Schrab et al. (2021, Appendix E.4), it is sufficient to bound the quantile $\widehat{q}_{1-\alpha}^{k,\infty}$ using wild bootstrapped statistics without finite approximation provided that $B_1 \ge \frac{3}{\alpha^2}\big(\log\big(\frac{8}{\beta}\big) + \alpha(1-\alpha)\big)$. Using the concentration bound for i.i.d. Rademacher chaos of de la Peña and Giné (1999, Corollary 3.2.6) as presented by Kim et al. (2022, Equation 39), there exists some constant $C > 0$ such that

$$\mathbb{P}_\epsilon\left(\left|\frac{1}{N(N-1)}\sum_{1\le i \ne j \le N}\epsilon_i\epsilon_j h_{p,k}(X_i,X_j)\right| \ge t \,\Big|\, \mathbb{X}_N\right) \le 2\exp\left(-\frac{CtN(N-1)}{\sqrt{\sum_{1\le i \ne j \le N} h_{p,k}(X_i,X_j)^2}}\right)$$

which gives

$$\widehat{q}_{1-\alpha}^{k,\infty} \le C\frac{\log(2/\alpha)}{N(N-1)}\sqrt{\sum_{1\le i \ne j \le N} h_{p,k}(X_i,X_j)^2}$$

for some different constant $C > 0$. Using Markov's inequality, for any $\delta \in (0,1)$, with probability at least $1 - \delta$ we have

$$\sum_{1 \leq i \neq j \leq N} h_{p,k}(X_i, X_j)^2 \leq \frac{1}{\delta} N(N-1) \mathbb{E}_{q,q} \big[ h_{p,k}(X,Y)^2 \big] = \frac{C_k N(N-1)}{\delta}$$

which gives

$$\widehat{q}_{1-\alpha}^{k,\infty} \leq C \frac{\log(2/\alpha)}{N(N-1)} \sqrt{\frac{C_k N(N-1)}{\delta}} = \frac{C}{\sqrt{\delta}} \frac{\log(2/\alpha)}{\sqrt{N(N-1)}} \sqrt{C_k} \leq C \log\left(\frac{1}{\alpha}\right) \frac{\sqrt{C_k}}{N}$$

since $\alpha \in (0, e^{-1})$, where in the last inequality $C > 0$ is a different constant depending on $\delta$. Using Dvoretzky–Kiefer–Wolfowitz inequality (Dvoretzky et al., 1956) as explained above, we deduce that

$$2\widehat{q}_{1-\alpha}^{k} \;\leq\; C \log\left(\frac{1}{\alpha}\right) \frac{\sqrt{C_k}}{N} \tag{12}$$

with arbitrarily high probability, for some constant $C > 0$. The same quantile bound also holds for the parametric bootstrap provided that $n/\sqrt{C_k} \geq \ln(1/\alpha)$ as derived by Albert et al. (2022, Appendix C.4.1). Their reasoning holds in our setting by replacing $1/\sqrt{\lambda_1 \cdots \lambda_p \mu_1 \cdots \mu_q}$ by $\sqrt{C_k}$, this is justified as for a kernel $k_\lambda$ with bandwidths $\lambda_1, \ldots, \lambda_d$ we have $\mathbb{E}_{q,q}[k_\lambda(X,Y)^2] \leq C/\lambda_1 \cdots \lambda_d$.

We now bound the variance term in Equation (10). Using the result of Albert et al. (2022, Equation 6), there exists $C > 0$ such that

$$\mathrm{var}_q\Big( \widehat{\mathrm{KSD}}_{p,k}^2(\mathbb{X}_N) \Big) \leq C \left( \frac{\sigma_1^2}{N} + \frac{\sigma_2^2}{N^2} \right)$$

where

$$\sigma_2^2 := \mathbb{E}_{q,q}\big[ h_{p,k}(X,Y)^2 \big] = C_k$$

and

$$\sigma_1^2 := \mathbb{E}_{Y \sim q}\left[ \Big( \mathbb{E}_{X \sim q}[h_{p,k}(X,Y)] \Big)^2 \right]$$

$$= \mathbb{E}_{Y \sim q}\left[ \Big( (T_{h_{p,k}} \psi)(Y) \Big)^2 \right]$$

$$= \int_{\mathbb{R}^d} \Big( (T_{h_{p,k}} \psi)(y) \Big)^2 q(y) \, \mathrm{d}y$$

$$\leq \|q\|_\infty \int_{\mathbb{R}^d} \Big( (T_{h_{p,k}} \psi)(y) \Big)^2 \, \mathrm{d}y$$

$$\leq M \big\| T_{h_{p,k}} \psi \big\|_2^2$$

since

$$(T_{h_{p,k}} \psi)(y) := \int_{\mathbb{R}^d} h_{p,k}(x,y) \psi(x) \, \mathrm{d}x$$

$$= \int_{\mathbb{R}^d} h_{p,k}(x,y) p(x) \, \mathrm{d}x - \int_{\mathbb{R}^d} h_{p,k}(x,y) q(x) \, \mathrm{d}x$$

$$= 0 - \int_{\mathbb{R}^d} h_{p,k}(x,y) q(x) \, \mathrm{d}x$$

$$= -\mathbb{E}_{X \sim q}[h_{p,k}(X,y)]$$

by Stein's identity. We deduce that

$$\mathrm{var}_q\Big( \widehat{\mathrm{KSD}}_{p,k}^2(\mathbb{X}_N) \Big) \leq C \left( \frac{\big\| T_{h_{p,k}} \psi \big\|_2^2}{N} + \frac{C_k}{N^2} \right)$$

for some different constant $C > 0$ depending on $M$ and $d$.

Using the classical inequalities $\sqrt{x+y} \le \sqrt{x} + \sqrt{y}$ and $2\sqrt{xy} \le x + y$ for $x, y > 0$, which are also considered in the works of Fromont et al. (2013), Albert et al. (2022) and Schrab et al. (2021), we obtain

$$
\begin{aligned}
2\sqrt{\frac{\mathrm{var}_q\left(\widehat{\mathrm{KSD}}^2_{p,k}(\mathbb{X}_N)\right)}{\beta}} &\le 2\sqrt{C\frac{\left\|T_{h_{p,k}}\psi\right\|_2^2}{\beta N} + C\frac{C_k}{\beta N^2}} \\
&\le 2\sqrt{\left\|T_{h_{p,k}}\psi\right\|_2^2\frac{C}{\beta N}} + 2\sqrt{C\frac{C_k}{\beta N^2}} \\
&\le \left\|T_{h_{p,k}}\psi\right\|_2^2 + \frac{C}{\beta N} + 2\sqrt{C}\frac{\sqrt{C_k}}{\sqrt{\beta}N} \\
&\le \left\|T_{h_{p,k}}\psi\right\|_2^2 + \left(C + 2\sqrt{C}\right)\frac{\sqrt{C_k}}{\beta N} \\
&\le \left\|T_{h_{p,k}}\psi\right\|_2^2 + C\log\left(\frac{1}{\alpha}\right)\frac{\sqrt{C_k}}{\beta N} \qquad (13)
\end{aligned}
$$

since $\alpha \in (0, e^{-1})$, $\beta \in (0, 1)$ and $C_k \ge 1$ by assumption, where the constant $C > 0$ is different on the last line. By considering the largest of the two constants in the quantile and variance bounds of Equations (12) and (13) multiplied by two, we obtain

$$
2\widehat{q}^k_{1-\alpha} + 2\sqrt{\frac{\mathrm{var}_q\left(\widehat{\mathrm{KSD}}^2_{p,k}(\mathbb{X}_N)\right)}{\beta}} \;\le\; \left\|T_{h_{p,k}}\psi\right\|_2^2 + C\log\left(\frac{1}{\alpha}\right)\frac{\sqrt{C_k}}{\beta N}.
$$

By combining this bound with the condition in Equation (11), we get that $\mathbb{P}_q\left(\Delta^k_\alpha(\mathbb{X}_N) = 0\right) \le \beta$ if

$$
\begin{aligned}
\|\psi\|_2^2 &\ge \left\|\psi - T_{h_{p,k}}\psi\right\|_2^2 - \left\|T_{h_{p,k}}\psi\right\|_2^2 + \left\|T_{h_{p,k}}\psi\right\|_2^2 + C\log\left(\frac{1}{\alpha}\right)\frac{\sqrt{C_k}}{\beta N}, \\
\|\psi\|_2^2 &\ge \left\|\psi - T_{h_{p,k}}\psi\right\|_2^2 + C\log\left(\frac{1}{\alpha}\right)\frac{\sqrt{C_k}}{\beta N},
\end{aligned}
$$

which concludes the proof.

### I.2   Proof of Proposition 3.2

Recall that the correction term in Equation (6) is defined as

$$
u_\alpha := \sup\left\{u \in \left(0, \min_{k \in \mathcal{K}} w_k^{-1}\right) : \frac{1}{B_2}\sum_{b=1}^{B_2}\mathbb{1}\left(\max_{k \in \mathcal{K}}\left(\widetilde{K}^b_k - \widehat{q}^k_{1-uw_k}\right) > 0\right) \le \alpha\right\}
$$

where $\widehat{q}^k_{1-uw_k} = \bar{K}_k^{\bullet\lceil(B_1+1)(1-uw_k)\rceil}$, as defined in Equation (3). Hence, we have

$$
\frac{1}{B_2}\sum_{b=1}^{B_2}\mathbb{1}\left(\max_{k \in \mathcal{K}}\left(\widetilde{K}^b_k - \widehat{q}^k_{1-u_\alpha w_k}\right) > 0\right) \;\le\; \alpha. \qquad (14)
$$

Recall that our estimator is $\widehat{\mathrm{KSD}}^2_{p,k}(\mathbb{X}_N)$ where $\mathbb{X}_N := (X_i)_{i=1}^N$ are drawn from $q$. Recall also from Equation (4), that for the parametric bootstrap each element $\widetilde{K}^b_k$ is computed by drawing new samples $(X'_i)_{i=1}^{N'}$ from the model $p$ and computing $\widehat{\mathrm{KSD}}^2_{p,k}\left((X'_i)_{i=1}^{N'}\right)$. Hence, at every sample size, under the null hypothesis $\mathcal{H}_0 : p = q$ the quantities $\widetilde{K}^b_k$ and $\widehat{\mathrm{KSD}}^2_{p,k}(\mathbb{X}_N)$ are identically distributed. For the wild bootstrap, as defined in Equation (5), Chwialkowski et al. (2014, 2016) show that, under $\mathcal{H}_0$, $\widetilde{K}^b_k$ and $\widehat{\mathrm{KSD}}^2_{p,k}(\mathbb{X}_N)$ have the same asymptotic distribution (details are presented in Appendix D).

Therefore, by taking the expectation in Equation (14), we obtain

$$
\begin{aligned}
\alpha \;\geq\; & \mathbb{P}_p\!\left(\max_{k \in \mathcal{K}}\left(\widehat{\mathrm{KSD}}^2_{p,k}(\mathbb{X}_N) - \widehat{q}^k_{1-u_\alpha w_k}\right) > 0\right)\\
=\; & \mathbb{P}_p\!\left(\widehat{\mathrm{KSD}}^2_{p,k}(\mathbb{X}_N) > \widehat{q}^k_{1-u_\alpha w_k} \ \text{ for some } \ k \in \mathcal{K}\right)\\
=\; & \mathbb{P}_p\!\left(\Delta^k_{u_\alpha w_k}(\mathbb{X}_N) = 1 \ \text{ for some } \ k \in \mathcal{K}\right)\\
=\; & \mathbb{P}_p\!\left(\Delta^{\mathcal{K}}_{\alpha}(\mathbb{X}_N) = 1\right)
\end{aligned}
$$

which holds non-asymptotically using the parametric bootstrap, and asymptotically using the wild bootstrap. Note that this is different from the two-sample case where using a wild bootstrap yields well-calibrated non-asymptotic levels (Schrab et al., 2021, Proposition 8).

The same reasoning holds by replacing $u_\alpha$ by any value $u \in (0, u_\alpha)$. In particular, it holds for the lower bound of the interval obtained by performing the bisection method to approximate the supremum in the definition of $u_\alpha$. This lower bound is the one we use in practice, as shown in Algorithm 1 with the step '$u_\alpha = u_{\min}$'. We have proved that the test with correction term $u_\alpha$ approximated with a bisection method also achieves the desired level $\alpha$.

## I.3   Proof of Theorem 3.3

As explained in Appendix I.2 and utilised in Algorithm 1, we use the lower bound $\widehat{u}_\alpha$ of the interval obtained by performing the bisection method to approximate the supremum in the definition of $u_\alpha$ in Equation (6). The assumptions $B_2 \geq \frac{8}{\alpha^2}\log\!\left(\frac{2}{\beta}\right)$ and $B_3 \geq \log_2\!\left(\frac{4}{\alpha}\min_{k \in \mathcal{K}} w_k^{-1}\right)$ ensure that $\widehat{u}_\alpha \geq \alpha/2$ as shown by Schrab et al. (2021, Appendix E.9).

The probability of type II error of $\Delta^{\mathcal{K}}_\alpha$ is

$$
\begin{aligned}
\mathbb{P}_q\!\left(\Delta^{\mathcal{K}}_{\alpha}(\mathbb{X}_N) = 0\right) \;=\; & \mathbb{P}_q\!\left(\Delta^k_{\widehat{u}_\alpha w_k}(\mathbb{X}_N) = 0 \ \text{ for all } \ k \in \mathcal{K}\right)\\
\leq\; & \mathbb{P}_q\!\left(\Delta^k_{\widehat{u}_\alpha w_k}(\mathbb{X}_N) = 0 \ \text{ for some } \ k \in \mathcal{K}\right)\\
\leq\; & \mathbb{P}_q\!\left(\Delta^k_{\alpha w_k/2}(\mathbb{X}_N) = 0 \ \text{ for some } \ k \in \mathcal{K}\right).
\end{aligned}
$$

To guarantee $\mathbb{P}_q\!\left(\Delta^{\mathcal{K}}_{\alpha}(\mathbb{X}_N) = 0\right) \leq \beta$, it is then sufficient to guarantee $\mathbb{P}_q\!\left(\Delta^{k^*}_{\alpha w_{k^*}/2}(\mathbb{X}_N) = 0\right) \leq \beta$ for some $k^* \in \mathcal{K}$ to be specified shortly in Equation (15). By assumption, we have

$$
B_1 \;\geq\; \left(\max_{k \in \mathcal{K}} w_k^{-2}\right)\frac{12}{\alpha^2}\left(\log\!\left(\frac{8}{\beta}\right) + \alpha(1-\alpha)\right).
$$

In order to apply Theorem 3.1 to $\Delta^k_{\alpha_k}$ with $\alpha_k := \alpha w_k/2$ for $k \in \mathcal{K}$, we need to ensure that the condition on $B_1$ of Theorem 3.1 is satisfied for all $k \in \mathcal{K}$, that is

$$
B_1 \;\geq\; \frac{3}{\alpha_k^2}\left(\log\!\left(\frac{8}{\beta}\right) + \alpha_k(1-\alpha_k)\right)
$$

for all $k \in \mathcal{K}$. Since $0 < \alpha_k < \alpha < e^{-1}$, we have $\alpha(1-\alpha) \geq \alpha_k(1-\alpha_k)$ for $k \in \mathcal{K}$. We also have

$$
\left(\max_{k \in \mathcal{K}} w_k^{-2}\right)\frac{12}{\alpha^2} \;\geq\; 3\left(\frac{2}{\alpha w_k}\right)^2 = \frac{3}{\alpha_k^2}
$$

for all $k \in \mathcal{K}$. We deduce that

$$
\begin{aligned}
B_1 \;\geq\; & \left(\max_{k \in \mathcal{K}} w_k^{-2}\right)\frac{12}{\alpha^2}\left(\log\!\left(\frac{8}{\beta}\right) + \alpha(1-\alpha)\right)\\
\geq\; & \frac{3}{\alpha_k^2}\left(\log\!\left(\frac{8}{\beta}\right) + \alpha_k(1-\alpha_k)\right)
\end{aligned}
$$

for all $k \in \mathcal{K}$, and so, applying Theorem 3.1 to $\Delta^k_{\alpha_k}$ for $k \in \mathcal{K}$ is justified. We obtain that $\mathbb{P}_q\!\left(\Delta^k_{\alpha w_k/2}(\mathbb{X}_N) = 0\right) \leq \beta$ if

$$
\|\psi\|_2^2 \;\geq\; \left\|\psi - T_{h_{p,k}}\psi\right\|_2^2 + C\log\!\left(\frac{2}{\alpha w_k}\right)\frac{\sqrt{C_k}}{\beta N},
$$

or, with a different constant $C > 0$, if

$$\|\psi\|_2^2 \geq \|\psi - T_{h_{p,k}}\psi\|_2^2 + C \log\left(\frac{1}{\alpha w_k}\right) \frac{\sqrt{C_k}}{\beta N}$$

since $\log\left(\frac{2}{\alpha w_k}\right) \leq (\log(2) + 1) \log\left(\frac{1}{\alpha w_k}\right)$ as $\alpha \in (0, e^{-1})$ and $w_k \in (0, 1)$. Now, let

$$k^* := \operatorname*{argmin}_{k \in \mathcal{K}} \left( \|\psi - T_{h_{p,k}}\psi\|_2^2 + C \log\left(\frac{1}{\alpha w_k}\right) \frac{\sqrt{C_k}}{\beta N} \right). \tag{15}$$

Finally, we have $\mathbb{P}_q\big(\Delta_\alpha^{\mathcal{K}}(\mathbb{X}_N) = 0\big) \leq \beta$ if $\mathbb{P}_q\Big(\Delta_{\alpha w_{k^*}/2}^{k^*}(\mathbb{X}_N) = 0\Big) \leq \beta$, that is, if

$$\|\psi\|_2^2 \geq \|\psi - T_{h_{p,k^*}}\psi\|_2^2 + C \log\left(\frac{1}{\alpha w_{k^*}}\right) \frac{\sqrt{C_{k^*}}}{\beta N}$$

$$= \min_{k \in \mathcal{K}} \left( \|\psi - T_{h_{p,k}}\psi\|_2^2 + C \log\left(\frac{1}{\alpha w_k}\right) \frac{\sqrt{C_k}}{\beta N} \right),$$

as desired.

## I.4   Proof of Theorem 3.5

Recall that we suppose that the following assumptions hold.

- The model density $p$ is strictly positive on its connected and compact support $S \subseteq \mathbb{R}^d$.
- The score function $\nabla \log p(x)$ is continuous and bounded on $S$.
- The support of the density $q$ is a connected and compact subset of $S$.
- The kernel used is a scaled Gaussian kernel $k_\lambda(x, y) := \lambda^{2-d} \exp\left(-\|x - y\|_2^2 / \lambda^2\right)$.

We introduce some notation. For some $c \in \mathbb{R}$, we write $\mathbf{c} := (c, \ldots, c) \in \mathbb{R}^d$. We write $(a_1, \ldots, a_d) \leq (b_1, \ldots, b_d)$ when $a_i \leq b_i$ for $i = 1, \ldots, d$. We use $C, C'$ to denote some generic constants which may change on different lines.

Note that, by properties of compactness on $\mathbb{R}^d$, there exists some $a > 0$ such that $S \subseteq \left[-\frac{a}{2}, \frac{a}{2}\right]^d$. Since the score function $\nabla \log p(x)$ is continuous and bounded on the compact set $S$, there exists some $c_1 > 0$ such that

$$|(\nabla \log p(x))_i| \leq c_1 \quad \text{for } i = 1, \ldots, d \text{ and for all } x \in S.$$

We work with a scaled Gaussian kernel with bandwidth $\lambda \leq 1$, defined as

$$k_\lambda(x, y) := \lambda^{2-d} \exp\left(-\frac{\|x - y\|_2^2}{\lambda^2}\right)$$

which satisfies

$$\nabla_x k_\lambda(x, y) = 2\lambda^{-d}(y - x) \exp\left(-\frac{\|x - y\|_2^2}{\lambda^2}\right) \leq 2\lambda^{-d}\mathbf{a} \exp\left(-\frac{\|x - y\|_2^2}{\lambda^2}\right),$$

$$\nabla_y k_\lambda(x, y) = 2\lambda^{-d}(x - y) \exp\left(-\frac{\|x - y\|_2^2}{\lambda^2}\right) \leq 2\lambda^{-d}\mathbf{a} \exp\left(-\frac{\|x - y\|_2^2}{\lambda^2}\right),$$

$$\sum_{i=1}^d \frac{\partial}{\partial x_i \partial y_i} k_\lambda(x, y) = \left(2d\lambda^{-d} - 4\lambda^{-2-d}\|x - y\|_2^2\right) \exp\left(-\frac{\|x - y\|_2^2}{\lambda^2}\right)$$

$$\leq 2d\lambda^{-d} \exp\left(-\frac{\|x - y\|_2^2}{\lambda^2}\right).$$

The Stein kernel associated to $k_\lambda$ satisfies

$$h_{p,k_\lambda}(x,y) := \left(\nabla \log p(x)^\top \nabla \log p(y)\right) k_\lambda(x,y) + \nabla \log p(y)^\top \nabla_x k_\lambda(x,y)$$

$$+ \nabla \log p(x)^\top \nabla_y k_\lambda(x,y) + \sum_{i=1}^{d} \frac{\partial}{\partial x_i \partial y_i} k_\lambda(x,y)$$

$$\leq 2\left(dc_1^2 + 2dc_1 a\lambda^{-d} + d\lambda^{-d}\right) \exp\left(-\frac{\|x-y\|_2^2}{\lambda^2}\right)$$

$$\leq C_0 \pi^{-d/2} \lambda^{-d} \exp\left(-\frac{\|x-y\|_2^2}{\lambda^2}\right)$$

since $\lambda \leq 1$, where the constant is $C_0 := 2\pi^{d/2}(dc_1^2 + 2dc_1 a + d)$. The Stein kernel can be upper bounded by a scaled Gaussian kernel on $S$. Writing

$$\bar{k}_\lambda(x,y) := \pi^{-d/2} \lambda^{-d} \exp\left(-\frac{\|x-y\|_2^2}{\lambda^2}\right),$$

which is of the form considered by Schrab et al. (2021, Section 3.1) with equal bandwidths for each dimension, we have shown that

$$h_{p,k_\lambda}(x,y) \leq C_0 \bar{k}_\lambda(x,y) \qquad \text{for all } x,y \in S. \tag{16}$$

Writing $\psi := p - q$, recall from Theorem 3.1 that a sufficient condition for control of the probability of type II error by $\beta$ is

$$\|\psi\|_2^2 \geq \left\|\psi - T_{h_{p,k_\lambda}}\psi\right\|_2^2 + C\log\left(\frac{1}{\alpha}\right)\frac{\sqrt{\mathbb{E}_{q,q}[h_{p,k_\lambda}(X,Y)^2]}}{\beta N}.$$

Using the upper bound

$$\mathbb{E}_{q,q}[h_{p,k_\lambda}(X,Y)^2] \leq C_0^2 \mathbb{E}_{q,q}\left[\bar{k}_\lambda(X,Y)^2\right] \leq C_0^2 \frac{M}{\lambda^d}$$

where the last inequality holds as in Schrab et al. (2021, Equation (22)), we obtain the sufficient condition

$$\|\psi\|_2^2 \geq \left\|\psi - T_{h_{p,k_\lambda}}\psi\right\|_2^2 + C\frac{\log(1/\alpha)}{\beta N \lambda^{d/2}} \tag{17}$$

where we have absorbed the term $C_0\sqrt{M}$ in the constant $C$.

We would like to upper bound the term $\left\|\psi - T_{h_{p,k_\lambda}}\psi\right\|_2^2$ by $\left\|\psi - T_{\bar{k}_\lambda}\psi\right\|_2^2$ but this is not possible using Equation (16). Instead, we can use the triangle inequality to get

$$\left\|\psi - T_{h_{p,k_\lambda}}\psi\right\|_2^2 \leq \left\|\psi - T_{\bar{k}_\lambda}\psi\right\|_2^2 + \left\|T_{\bar{k}_\lambda}\psi - T_{h_{p,k_\lambda}}\psi\right\|_2^2$$

$$\leq \left\|\psi - T_{\bar{k}_\lambda}\psi\right\|_2^2 + \left(\left\|T_{\bar{k}_\lambda}\psi\right\|_2 + \left\|T_{h_{p,k_\lambda}}\psi\right\|_2\right)^2$$

$$\leq \left\|\psi - T_{\bar{k}_\lambda}\psi\right\|_2^2 + 2\left(\left\|T_{\bar{k}_\lambda}\psi\right\|_2^2 + \left\|T_{h_{p,k_\lambda}}\psi\right\|_2^2\right)$$

$$\leq \left\|\psi - T_{\bar{k}_\lambda}\psi\right\|_2^2 + 2(C_0^2 + 1)\left\|T_{\bar{k}_\lambda}\psi\right\|_2^2$$

since

$$\left(T_{h_{p,k_\lambda}}\psi\right)(y) = \int_S h_{p,k_\lambda}(x,y)\psi(x)\,\mathrm{d}x \leq C_0 \int_S \bar{k}_\lambda(x,y)\psi(x)\,\mathrm{d}x = C_0\left(T_{\bar{k}_\lambda}\psi\right)(y)$$

for all $y \in S$.

Recall that we assume that $\psi \in \mathcal{S}_d^{s,t}(R,L)$ for some $L$ to be determined, in particular $\psi \in \mathcal{S}_d^s(R)$.

For the term $\left\|\psi - T_{\bar{k}_\lambda}\psi\right\|_2^2$, we use the fact that $\psi \in \mathcal{S}_d^s(R)$. The term $\left\|\psi - T_{\bar{k}_\lambda}\psi\right\|_2^2$ can then be upper bounded exactly as done by Schrab et al. (2021, Appendix E.6) with the difference that we

choose $\widetilde{t} > 0$ such that $S_1 < 0.5$ rather than $S_1 < 1$ ($S_1$ is defined in Schrab et al. (2021, Appendix E.6)). Following their reasoning, since $\psi \in \mathcal{S}_d^s(R)$, we obtain that there exists some $S_1 \in (0, 0.5)$ and some constant $C > 0$ (depending on $d$, $s$ and $R$) such that

$$\left\|\psi - T_{\bar{k}_\lambda}\psi\right\|_2^2 \leq S_1^2\|\psi\|_2^2 + C\lambda^{2s} \leq \frac{1}{4}\|\psi\|_2^2 + C\lambda^{2s}.$$

To upper bound the term $\left\|T_{\bar{k}_\lambda}\psi\right\|_2^2$, we modify the reasoning of Schrab et al. (2021, Appendix E.6) used for the term $\left\|\psi - T_{\bar{k}_\lambda}\psi\right\|_2^2$, and utilise the restricted Sobolev ball regularity assumption. First, note that since $\bar{k}_\lambda$ is translation-invariant, $T_{\bar{k}_\lambda}$ is a convolution as

$$\left(T_{\bar{k}_\lambda}\psi\right)(y) = \int_S \bar{k}_\lambda(x, y)\psi(x)\,\mathrm{d}x = \int_S \varphi_\lambda(x - y)\psi(x)\,\mathrm{d}x = (\psi * \varphi_\lambda)(y)$$

where

$$\varphi_\lambda(u) := \prod_{i=1}^d \lambda^{-1}K\left(\frac{u_i}{\lambda}\right), \qquad \text{and} \qquad K(u) := \pi^{-1/2}\exp(-u^2).$$

By properties of Fourier transforms, we have $\widehat{\varphi_\lambda}(\xi) = \prod_{i=1}^d \widehat{K}(\lambda\xi_i)$ for $\xi \in \mathbb{R}^d$. Note that

$$\left|\prod_{i=1}^d \widehat{K}(\xi_i)\right| \leq \prod_{i=1}^d \int_\mathbb{R}|K(x)e^{-ix\xi_i}|\,\mathrm{d}x = \prod_{i=1}^d \int_\mathbb{R}|K(x)|\,\mathrm{d}x = 1.$$

For some $L$ to be determined, by assumption, we have $\psi \in \mathcal{S}_d^{s,t}(R, L)$, so

$$\int_{\|\xi\|_2 \leq t}\left|\widehat{\psi}(\xi)\right|^2\mathrm{d}\xi \leq \frac{1}{L}\int_{\mathbb{R}^d}\left|\widehat{\psi}(\xi)\right|^2\mathrm{d}\xi = \frac{\|\widehat{\psi}\|_2^2}{L}.$$

For $s > 0$, define

$$T_s := \sup_{\|\xi\|_2 > t}\frac{\left|\prod_{i=1}^d \widehat{K}(\xi_i)\right|}{\|\xi\|_2^s} \leq \frac{1}{t^s} < \infty.$$

Then, using Plancherel's Theorem as in Schrab et al. (2021, Appendix E.6), we obtain

$$\begin{aligned}
&(2\pi)^d\left\|T_{\bar{k}_\lambda}\psi\right\|_2^2 \\
&= (2\pi)^d\|\psi * \varphi_\lambda\|_2^2 \\
&= \left\|\widehat{\varphi_\lambda}\widehat{\psi}\right\|_2^2 \\
&= \int_{\mathbb{R}^d}|\widehat{\varphi_\lambda}(\xi)|^2\left|\widehat{\psi}(\xi)\right|^2\mathrm{d}\xi \\
&= \int_{\mathbb{R}^d}\left|\prod_{i=1}^d \widehat{K}(\lambda\xi_i)\right|^2\left|\widehat{\psi}(\xi)\right|^2\mathrm{d}\xi \\
&= \int_{\|\xi\|_2 \leq t}\left|\prod_{i=1}^d \widehat{K}(\lambda\xi_i)\right|^2\left|\widehat{\psi}(\xi)\right|^2\mathrm{d}\xi + \int_{\|\xi\|_2 > t}\left|\prod_{i=1}^d \widehat{K}(\lambda\xi_i)\right|^2\left|\widehat{\psi}(\xi)\right|^2\mathrm{d}\xi \\
&\leq \int_{\|\xi\|_2 \leq t}\left|\widehat{\psi}(\xi)\right|^2\mathrm{d}\xi + T_s^2\int_{\|\xi\|_2 > t}\|\lambda\xi\|_2^{2s}\left|\widehat{\psi}(\xi)\right|^2\mathrm{d}\xi \\
&\leq \frac{1}{L}\|\widehat{\psi}\|_2^2 + \lambda^{2s}T_s^2\int_{\mathbb{R}^d}\|\xi\|_2^{2s}\left|\widehat{\psi}(\xi)\right|^2\mathrm{d}\xi \\
&\leq \frac{(2\pi)^d}{L}\|\psi\|_2^2 + \lambda^{2s}T_s^2(2\pi)^dR^2.
\end{aligned}$$

Combining those results, the upper bound on $\left\|\psi - T_{h_{p,k_\lambda}}\psi\right\|_2^2$ becomes

$$\left\|\psi - T_{h_{p,k_\lambda}}\psi\right\|_2^2 \leq \left\|\psi - T_{\bar{k}_\lambda}\psi\right\|_2^2 + 2(C_0^2 + 1)\left\|T_{\bar{k}_\lambda}\psi\right\|_2^2$$

$$\leq \left(\frac{1}{4}\|\psi\|_2^2 + C\lambda^{2s}\right) + \left(\frac{2(C_0^2 + 1)}{L}\|\psi\|_2^2 + C'\lambda^{2s}\right)$$

$$= \left(\frac{1}{4}\|\psi\|_2^2 + C\lambda^{2s}\right) + \left(\frac{1}{4}\|\psi\|_2^2 + C'\lambda^{2s}\right)$$

$$\leq \frac{1}{2}\|\psi\|_2^2 + C\lambda^{2s},$$

where we let $L := 8(C_0^2 + 1)$. The condition in Equation (17) then becomes

$$\|\psi\|_2^2 \geq \frac{1}{2}\|\psi\|_2^2 + C\lambda^{2s} + C'\frac{\log(1/\alpha)}{\beta N \lambda^{d/2}}$$

which gives

$$\|\psi\|_2^2 \geq C\left(\lambda^{2s} + \frac{\log(1/\alpha)}{\beta N \lambda^{d/2}}\right). \tag{18}$$

Having proved the power guaranteeing condition in Equation (18), we can then derive rates for the KSD and KSDAGG tests as similarly done by Schrab et al. (2021). Set $\lambda := N^{-2/(4s+d)}$ to get the condition

$$\|\psi\|_2^2 \geq C\left(N^{-4s/(4s+d)} + N^{-1}N^{d/(4s+d)}\right) = CN^{-4s/(4s+d)}$$

which is the minimax rate over (unrestricted) Sobolev balls $\mathcal{S}_d^s(R)$ (Li and Yuan, 2019; Balasubramanian et al., 2021; Albert et al., 2022; Schrab et al., 2021).

In practice, the smoothness parameter of the Sobolev ball is not known, so we cannot set $\lambda := N^{-2/(4s+d)}$ (i.e. it cannot be implemented). Instead, we can use our aggregated test KSDAGG. Adapting the proof of Theorem 3.3 with the single test power condition Equation (18), we obtain the aggregated test power condition

$$\|\psi\|_2^2 \geq C\min_{\lambda \in \Lambda}\left(\lambda^{2s} + \log\left(\frac{1}{\alpha w_\lambda}\right)\frac{1}{\beta N \lambda^{d/2}}\right). \tag{19}$$

Similarly to Schrab et al. (2021, Corollary 10), consider

$$\Lambda := \left\{2^{-\ell} : \ell \in \left\{1, \dots, \left\lceil\frac{2}{d}\log_2\left(\frac{N}{\ln(\ln(N))}\right)\right\rceil\right\}\right\}, \qquad w_\lambda := \frac{6}{\pi^2 \ell^2}.$$

Let

$$\ell_* := \left\lceil\frac{2}{4s+d}\log_2\left(\frac{N}{\ln(\ln(N))}\right)\right\rceil \leq \left\lceil\frac{2}{d}\log_2\left(\frac{N}{\ln(\ln(N))}\right)\right\rceil.$$

The bandwidth $\lambda_* := 2^{-\ell_*} \in \Lambda$ satisfies

$$\ln\left(\frac{1}{w_{\lambda_*}}\right) \leq C\ln(\ell_*) \leq C\ln(\ln(N))$$

as $w_{\lambda_*} := 6\pi^{-2}(\ell_*)^{-2}$, and

$$\frac{1}{2}\left(\frac{N}{\ln(\ln(N))}\right)^{-2/(4s+d)} \leq \lambda_* \leq \left(\frac{N}{\ln(\ln(N))}\right)^{-2/(4s+d)}.$$

We get

$$\lambda_*^{2s} \leq \left(\frac{N}{\ln(\ln(N))}\right)^{-4s/(4s+d)}$$

and

$$\log\left(\frac{1}{\alpha w_{\lambda_*}}\right)\frac{1}{\beta N \lambda_*^{d/2}} \leq C\left(\frac{N}{\ln(\ln(N))}\right)^{-1}\left(\frac{N}{\ln(\ln(N))}\right)^{d/(4s+d)} \leq C\left(\frac{N}{\ln(\ln(N))}\right)^{-4s/(4s+d)}.$$

The KSDAGG power condition of Equation (19) then becomes

$$\|\psi\|_2^2 \geq C\left(\frac{N}{\ln(\ln(N))}\right)^{-4s/(4s+d)},$$

which concludes the proof.