# OpenReview forum: "KSD Aggregated Goodness-of-fit Test"
_NeurIPS.cc/2022/Conference — NeurIPS 2022 Accept_

### Official Review · Reviewer_ZqFj · 2022-07-07

**Rating:** 6
**Confidence:** 4
**Soundness:** 3 good
**Presentation:** 2 fair
**Contribution:** 2 fair

**Summary:**

This paper proposed a new goodness-of-fit (GOF) testing method based on KSD, which sidesteps the challenge of selecting a single kernel for the test. Instead, the proposed method, KSDAGG, can aggregate tests with a collection of kernels so that it maximises the test power over these kernels. This is achieved by performing a GOF test with each kernel in the collection and rejecting the null hypothesis if any one of the tests rejects it. To ensure that the KSDAGG can still control the user-defined type-I error, the author proposed a bisection algorithm to select the proper test interval for each kernel in the collection. Theoretically, the author showed a condition for a uniform separation rate so that the KSDAGG can also control the type-II error. This condition shows that KSDAGG achieves the smallest uniform separation rate of the collection.

Empirically, the author applied KSDAGG to the kernel bandwidth selection problem and compared it with kernel selection baselines including data splitting, median heuristic and kernel selection with extra data. It shows that it achieves similar test power as the kernel selection with extra data on synthetic data and the best performance for MNIST normalising flow.

**Questions:**

1. Like I have mentioned, the author should consider elaborating more on their novel contributions: why the extension to GOF is not trivial? Is it because of the derivation of the bound for uniform separation rate?

2. Another concern is the computational cost. It seems that with the aggregation trick, it performs a single test for each kernel in the collection. This means with a large collection (or the number of bootstrap samples), the computational cost can be much larger than a single test (with kernel selection). Can the author provide the performances with wall clock time? I am curious to see the trade-off.

3. For a fixed set of weight $w_k$ and level $\alpha$, do we have a "fixed" choice of $u_\alpha$? Clearly, from the bisection algorithm, we won't have exactly the same $u_\alpha$, but they are closed to each other. So here "fixed" means a very distinct set of $u_\alpha$. Another question is can we select the best kernel using the KSDAGG. For example, if one kernel in the collection rejects the null hypothesis, does it mean this is the best kernel?

4. The introduction of KSD can be improved. For example, instead of directly introducing KSD and the complicated Stein kernel and Stein identity. Maybe consider adding a bit of intuition on Stein discrepancy. In line 87, consider adding one sentence to explain what is the consistency of the Stein test? Also for corollary 3.4, it seems that this is identical to theorem 3.3, which doesn't need to be re-introduced. Saving this space allows the author to expand a bit more on the background.

5. For the synthetic experiment, why KSDAGG is better than KSD split extra data in RBM? It will be interesting to see the optimised bandwidth and the bandwidth of one that rejects the null in the collection.

**Limitations:**

This paper does not have any negative societal impact. But this paper does not seem to discuss its limitations enough. For example, the trade-off between high computational cost and test power. It seems that this method cannot be applied to kernels that require training like the deep kernel.

**Strengths And Weaknesses:**

**Strength**:
This paper takes an alternative point of view on the kernel selection problem, where it extends the recently proposed two-sample test aggregation method to goodness-of-fit with Stein discrepancy.
The presentation clarity is reasonable but can be much improved. In terms of significance, I think the targeted community is generally limited but it is a nice addition to it.

**Weakness**:
One of my confusions is its contribution. The author mentioned that the aggregation trick is not novel, which has already been proposed in the two-sample test. Thus, I fail to see the main contributions of KSDAGG. I saw the contribution section but I suggest the author summarise what are the key points and why they are novel. For example, why the extension to the GOF test is non-trivial (expand a bit on the challenge mentioned in related work?), what method do I use to solve this non-trivial extension, etc. So that the contribution and novelty are clearer.

In terms of clarity, the background material to KSD is a bit dense. Since I have worked with the Stein method before, it is clear to me, but I can see that for a more general audience, it can be a bit difficult to understand. I suggest considering adding a more detailed background section in the appendix?

---

> ### Author Response · Authors · 2022-08-02
> **Review response**
>
> We thank reviewer ZqFj for their questions and suggestions. We hope the discussion in the response to all reviewers (see box "Comments to all reviewers" above) have addressed the concerns expressed by the reviewer. We welcome the suggestion of adding a more detailed background section in the appendix, which will be reflected in the final version.
>
> **Q1:** See response to all reviewers above.
>
> **Q2**: The time complexity of KSDAgg is provided in Algorithm 1. Indeed it grows linearly with the number of kernels, quadratically with the sample size, and linearly with the number of bootstrap samples.
>
> MNIST Normalizing Flow (average of 10 run times with wild bootstrap)
>
> $n=100$, KSDAgg: 0.0372s, Median: 0.0046s, Split: 0.0217s, Split Extra: 0.0233s
>
> $n=200$, KSDAgg: 0.0841s, Median: 0.0099s, Split: 0.0637s, Split Extra: 0.0698s,
>
> $n=300$, KSDAgg: 0.1622s, Median: 0.0198s, Split: 0.1320s, Split Extra: 0.1452s,
>
> $n=400$, KSDAgg: 0.2759s, Median: 0.0336s, Split: 0.2298s, Split Extra: 0.2534s
>
> $n=500$, KSDAgg: 0.4212s, Median: 0.0510s, Split: 0.3585s, Split Extra: 0.3947s
>
> The time complexity of KSDAgg is $\mathcal{O}(\mid \Lambda\mid (B_1+B_2) N^2)$ and the one for median KSD is $\mathcal{O}(B_1 N^2)$ where $\mid \Lambda\mid = 21$ and $B_1=B_2=500$. While KSDAgg takes roughly 10 times longer to run than KSD median, we could have expected a larger difference looking at the time complexities. This can be explained by the fact that there are two major time-consuming steps: (i) computing the kernel matrices and (ii) computing the wild bootstrap samples. While (i) has complexity  $\mathcal{O}(N^2)$ and (ii) complexity $\mathcal{O}(BN^2 + NB^2)$, the constant for step (i) is much larger than the one for step (ii) (which is some matrix multiplication). Note that for KSDAgg to compute the $\mid \Lambda\mid$ kernel matrix, we need to compute the matrix of pairwise distances only once.
>
> When splitting the data, the computationally expensive step is to select the bandwidth. All the $\mid \Lambda\mid$ kernel matrices need to be computed as for KSDAgg, which is the expensive step (i). The split test runs only slightly faster than the split extra test, it runs faster than KSDAgg but their run times are of the same order of magnitude.
>
> **Q3:** Given some fixed weights and level there does not exist a single $u_\alpha$ associated to them. This is the strength of this multiple testing correction. First, note that it can be shown that $u_\alpha \geq \alpha$, this means that this multiple testing strategy is always as powerful as using a Bonferroni correction. Essentially, the Bonferroni correction comes from a union bound, the multiple testing strategy aims to tighten the bound. Here are two extreme examples to provide intuition about the multiple testing strategy. First, assume that $\ell$ events are all disjoints, then the union bound is tight and both Bonferroni and the method we use yield adjusted levels $\alpha/\ell$. Second, assume that all events are the same (or almost the same), then the Bonferroni correction still yields adjusted levels $\alpha/\ell$, while multiple testing strategy will give 'adjusted' levels $\alpha$.
>
> When KSDAgg rejects the null, we can check which specific kernels rejected the adjusted tests: this provides the kernels/bandwidths which are well-adapted to the problem, i.e. the "best" selection of bandwidths is naturally returned as a side-effect of the test (without requiring data splitting). This also contributes to interpretability of the resulting test, for instance if different kernels prioritise different features.
>
> **Q4:** We appreciate the reviewer's suggestions to improve the clarity of the paper. We will follow those suggestions for the final version of this paper.
>
> **Q5:** We assume the reviewer is asking about the MNIST Normalizing Flow experiment where KSDAgg obtains high power while KSD split extra does not. In this setting, the median bandwidth is on average 2437. The collection consists of the median bandwidth scaled by $2^i$ for $i=-20,\dots,0$. When KSDAgg rejects the null hypothesis, the smallest bandwidth (among others) rejects the single test with adjusted level, note that this bandwidth is $2^{-20}\cdot 2437 \approx 0.002$. The bandwidth selected by the by split extra is the largest bandwidth of the collection, that is the median bandwidth (roughly 2437). The proxy used for the bandwidth selection is to maximize asymptotic power! In this high-dimensional setting ($d=784$) with sample sizes smaller than 500, we are clearly not in the asymptotic regime, which explains the low power obtained by KSD split extra.

---

> > ### Comment · Reviewer_ZqFj · 2022-08-05
> > **Reply to author response**
> >
> > I really appreciate the author's detailed response, which has addressed most of my concerns. The explanation of Q3 is quite useful to understand this method. Maybe consider adding this discussion to the main paper?
> >
> > If I understood correctly, for Q5, you said the KSDAgg select the bandwidth 0.002 but split extra select 2437? Why there is such a big difference even with limited data?
> >
> > I have increased my score after reading the author's response.

---

> > > ### Author Response · Authors · 2022-08-09
> > > **Response to Reviewer ZqFj**
> > >
> > > We thank reviewer ZqFj for their reply, and for increasing their score!
> > >
> > > We will follow their suggestion and include a discussion of the advantages of the multiple testing strategy used against the classical Bonferroni correction.
> > >
> > > Yes, KSDAgg selects the bandwidth 0.002 and split extra selects 2437. Split extra selects the bandwidth which maximizes
> > > $$
> > > \widehat{\textrm{KSD}}^{p,\lambda}  / \widehat\sigma_{\lambda}
> > > $$
> > > where $\widehat\sigma_{\lambda}^2$ is a regularised positive estimator of the asymptotic variance of $\widehat{\textrm{KSD}}_{p,\lambda}$ under  the alternative, as explained lines 195/196. Maximizing this criterion is equivalent to maximizing asymptotic power, as was shown by D. J. Sutherland et al., 2017 (Generative Models and Model Criticism via Optimized Maximum Mean Discrepancy) for the MMD; the same result holds straightforwardly for the KSD due to similar asymptotic properties. However, this criterion only maximizes **asymptotic** power and has no guarantee when using limited data. In our high-dimensional setting ($d=784$) with small sample size $N\leq 500$, the asymptotic regime is clearly not reached, and the criterion used for bandwidth selection does not maximize power in this **non-asymptotic** setting. So, even though split extra has access to some extra data, it does not have an accurate criterion to select the bandwidth and ends up selecting the largest bandwidth, which is not well-adapted to the problem. This explains why such a big difference in bandwidths is observed. We will clarify this finding in the main paper.
> > >
> > > We also point the reviewer to our general comment **For all reviewers: Updated version of the paper (Appendix D)**, we have now updated the paper showing minimax optimality and adaptivity of KSDAgg in the setting in which the densities are compactly supported.

---

### Official Review · Reviewer_CDhe · 2022-07-11

**Rating:** 7
**Confidence:** 4
**Soundness:** 4 excellent
**Presentation:** 4 excellent
**Contribution:** 2 fair

**Summary:**

In this work, the authors propose a natural extension to the existing KSD goodness of fit tests, but allowing aggregate testing over multiple kernels (in particular, multiple kernel hyper-parameters) without requiring strategies such as data-splitting etc.    They achieve this by rescaling the critical value of each individual test by a positive weight, summing up to <= 1.  When the weights are equal to 1/N (where N is the number of tests), this is essentially a Bonferroni-type correction for aggregating finitely many independent tests.

The authors provide an algorithm for performing this multiple testing protocol, based on wild-bootstrap or parametric bootstrap.  They then establish control over type II error through a uniform separation rate (USR) argument.

Many of the arguments are similar to what is proposed in the preprint [Schrab, A., Kim, I., Albert, M., Laurent, B., Guedj, B., and Gretton, A. (2021). MMD aggregated two-sample test], which considers the similar case of MMD testing.  But in that situation things are more straightforward since terms within the USR argument can be bounded explicitly in terms of best approximation rates for densities with a given regularity, yielding minimax optimal rates, etc.     In this setting it is far less straightforward to obtain similar bounds.

**Questions:**

1. Do the authors have any intuition on the behaviour of the terms in the uniform separate rates established for the aggregate test in terms of dimension?  Even just heuristics or a case study on a Gaussian Stein kernel would be of interest.
2. Do the authors have any recommendations on the choice of the weights?  This seems arbitrary and not really addressed anywhere in this work?  Is there any scenario where taking the sum to be < 1 make sense?
3.  One often wishes to select a kernel over an infinite continuum family of parameters, and it's sometimes less obvious how to discretise this.  Are the authors aware of any strategies which would enable them to generalise to this setting.

**Limitations:**

I am satisfied with how the authors addressed the limitations of this work.

**Strengths And Weaknesses:**

Strengths:
* (Quality + Clarity) It is very well written.  The arguments are presented cleanly which made following the proofs quite easy.
* (Significance) It tackles an important challenge which provides a principled alternative to the median heuristic or data-splitting approaches to addressing this problem.

Weaknesses:
 * (Originality) It is a bit incremental, and builds very obviously on previous work which tackled aggregate tests for MMD in the same light.
 * (Significance) While the uniform separation rates established in this paper are of theoretical interest, they aren't actionable in any manner due to the complexity of the Stein kernel.

---

> ### Author Response · Authors · 2022-08-02
> **Review response**
>
> We warmly thank reviewer CDhe for praising the soundness, clarity and significance of our work!
>
> Please see also the response to all reviewers above.
>
> **Comment**: *When the weights are equal to $1/N$ (where $N$ is the number of tests), this is essentially a Bonferroni-type correction for aggregating finitely many independent tests.* Actually, this is always at least as powerful as a Bonferroni-type correction. It can be shown that when the weights are $1/N$, then the correction $u \geq \alpha$, which means that the test will always reject the null when Bonferroni correction would reject it. Essentially, Bonferroni correction uses a loose union bound argument and the method used is trying the tighten this loose upper bound. For an extreme example illustrating the difference, imagine that all the kernels in the collection are the same, then a Bonferroni correction would be $\alpha/N$ while the method used would give level $\alpha$ as there is nothing to correct for since all the kernels are the same.
>
> **Q1**: The behaviour of the statistic as a function of dimension is subtle: in particular, increasing dimension might make the problem harder *or* easier. To illustrate with earlier work in the two-sample setting: when two multivariate Gaussians  differ along a single dimension by a fixed amount, then increasing the number of dimensions will make the problem harder, and test power will decrease. When two multivariate Gaussians with the same mean differ in variance across all their dimensions, then test power will increase as evidence accumulates with increasing dimension. See Gretton et al (2012a) Figure 5 for both cases.  We propose to add a study when p and q are both multivariate Gaussian, covering both scenarios, and to investigate test power with increasing dimension, for the final version, to verify similar behaviour occurs for KSD.
>
>
>
> **Q2:** Without any prior knowledge (which is often the case in practice), we recommend using uniform weights since we do not expect particular bandwidths to be better-suited than others. If the user has some prior knowledge of which bandwidths would be better for the task considered, then higher weights on those bandwidths can be used. Allowing for weights whose sum is strictly smaller than 1 is only for convenience of being able to add a new bandwidth with a new weight without changing the previous weights (for examples with weights $\frac{6}{\pi^2\ell^2}$ for $\ell\in\mathbb{N}\setminus\{0\}$). Multiplying all the weights by a constant simply results in dividing the correction $u_\alpha$ defined in Equation (5) by the same constant. This means that the product $u_\alpha w_\lambda$ remains the same, and hence the definition of the aggregated test is not affected by this sample. For simplicity, in practice, we use weights whose sum is equal to 1. We will add a discussion of the choice of weights in the final version.
>
> **Q3:** Extension to the case of a continuous collection of kernels (indexed by the bandwidth parameter on the positive real line) is a direction for future work, however, this extension is far from trivial. To the best of our knowledge, this has currently never been done  without data splitting. Our tests retain high power even with large collections of kernels, and our method also allows to aggregate multiple kernels (Gaussian, Laplace, IMQ, Matérn, etc.) with different parameters, but the extension to a continuous parametrization remains a challenge.

---

> > ### Comment · Reviewer_CDhe · 2022-08-08
> > **Response from Reviewer**
> >
> > Thank you for your detailed replies and clarifications.  I'm happy with the clarifications you have made, particularly with regards the Bonferroni query.  Based on your replies to me and to other authors I will increase my score.

---

> > > ### Author Response · Authors · 2022-08-09
> > > **Response to Reviewer CDhe**
> > >
> > > We thank reviewer CDhe for their response and for increasing their score!
> > >
> > > Regarding Q1, we have now provided an updated version of the paper where we consider in Appendix D the setting in which the densities have compact support, this includes for example d-dimensional isotropic Gaussians truncated to some compact subset of $\mathbb{R}^d$. Please see **For all reviewers: Updated version of the paper (Appendix D)** and to the updated paper for details.
> > >
> > > Q1 asked about the behaviour of the two terms in the condition of Theorem 3.1 with respect to dimension.
> > > Recall that the first term $(1)=\|\psi-T_{h_{p,k}} \psi\|_2^2$ indicates the size of the effect of the Stein operator on the difference in densities $\psi = p-q$, and is a measure of distance from the null (where this quantity is zero), and that the second term $(2)=\log(1/\alpha) \frac{\sqrt{C_k}}{\beta N}$ is obtained from upper bounding the variance of the KSD $U$-statistic and the quantile as explained in the proof of Theorem 3.1.
> > >
> > > Our analysis in Appendix D shows the dependence of the two terms on the dimension with respect to the bandwidth. As shown in Equation (16) with bandwidth $\lambda\leq 1$, the first term $(1)$ gives $\lambda^{2s}$ where $s$ is the smoothness parameter of the Sobolev ball while term $(2)$ gives $\lambda^{-d/2}$. For the first one, the bandwidth component has no dependence on dimension, due to the Sobolev assumption. The second term increases with dimension as $\lambda\leq 1$. This reasoning holds for bandwidths independent of the dimension. In order to obtain the minimax rate we need to set the bandwidth depending on the dimension, (i.e. $\lambda=N^{-2/(4s+d)}$).

---

### Official Review · Reviewer_ys6t · 2022-07-12

**Rating:** 6
**Confidence:** 4
**Soundness:** 2 fair
**Presentation:** 3 good
**Contribution:** 3 good

**Summary:**

This work provides guarantees for goodness-of-fit tests (whether a given set of samples comes from a given null distribution) based on kernel stein discrepancies (KSD). They construct a test that allows for kernel selection to maximize power by using all data rather than using data split while maintaining the type I error level. In particular, the work extends the idea of aggregated tests already used for the two-sample problems via MMDs, and the independence problem via HSIC, to the goodness-of-fit setting via KSD. To estimate the test thresholds, they make use of a Monte Carlo strategy based on (a) parametric bootstrap (which requires samples from the null distribution p) and provides a non-asymptotic level, and power; and (b) wild bootstrap that provides the same guarantees asymptotically.

**Questions:**

1. Can the authors clarify what the constant C depends on in Theorem 3.1?

2. Corollary 3.4 has a logarithmic inflation factor in front of the sample size N (in the second term). If one uses an equal sample split to do kernel selection first and then test, isn't that factor just 2? If yes, then is KSDAGG better than sample split only if log(#kernels) <=2? If not, it would help to clarify what the rate for sample split is known to be.

3.  Given the non-transparent nature of the Stein operator, can the authors provide some examples of q's that satisfy theorem 3.1 / 3.3 for a given simple distribution p, say Gaussian distribution? The authors make a comment in l 175--176, but it would be helpful to provide concrete examples (also relevant for comment 2).

4. In Proposition 3.2, is there no requirement on B1, B2, and B3?

5. Can the authors state a mathematically precise asymptotic result for wild bootstrap? (Like how is the limit for N to infinity taken?)

6. Is there no setting, where the KSD median would be competitive? It would be helpful to comment on the limitations of KSDAGG, namely when does KSDAGG either perform poorly or not better than the other sensible benchmarks.

7. Do we really need to know the model density? Or just the gradient of log density, and a sampler of p for parametric bootstrap? (OPTIONAL: What if the sampler is approximate?)

8. Given the unknown nature of constants in the uniform separation rate, can the authors comment on how the power degrades when the separation deteriorates from the mentioned lower bound?

Minor comments:

- To my understanding from the proof, Theorem 3.3 is effectively a corollary of Theorem 3.1 and a union bound; it would be helpful to provide that comment if true?


**Limitations:**

See questions.

**Strengths And Weaknesses:**

+ Their Theorem 3.1 characterizes the uniform separation rate between the null and the alternative densities so as to assert a certain power for a given test, and Theorem 3.3 extends it to multiple aggregated tests. The authors provide several numerical experiments that showcase that KSDAGG performs better than some other benchmarks.

+ Writing of the paper is very good, and the authors cover a fair amount of related work!

- This work applies only to those settings where model density is known.

For more comments, see questions.

---

> ### Author Response · Authors · 2022-08-02
> **Review response**
>
> We thank reviewer ys6t for summarising the strengths of the paper, and for their kind words on the clarity and quality of writing.
>
> Please also see response to all reviewers above.
>
> **Q1:** The constant $C$ in Theorem 3.1 depends only on $M$ and on $d$. We will add this to the statements of the theorems.
>
> **Q2:** In Theorem 3.1, the dependence is $\log(1/\alpha)$. This gives rise in Corollary 3.4 to a dependence $\log(1/\alpha w_\lambda)$. Since we require $\sum_\lambda w_\lambda \leq 1$, the weights for a collection $\{\lambda_1,\dots,\lambda_L\}$ are often defined as $w_\ell \coloneqq \frac{6}{\pi^2 \ell^2}$ for $\ell=1,\dots,L$ so that the series converge. In this case, we would get $\log(1/\alpha) \leq C \log(\ell) \leq C \log(L)$. It is not directly clear what the rate would be for sample split since this would depend on how the kernel/bandwidth is selected.
>
> **Q3:** Thank you for the suggestion - we will demonstrate this principle for the Gaussian case, as suggested (in this case, the Stein operator and relevant operations are computable in closed form). The alternatives we will demonstrate are "same variance different mean" and "same mean different variance," to illustrate local departures from the null. We will consider in particular the behaviour of the statistic as a function of dimension. See also the reply to reviewer CDhe question 1.
>
>
> **Q4:** The probability of type I error is always controlled by $\alpha$. Intuitively, if $B_1$, $B_2$ and $B_3$ are 'too small' then the probability of type I error will strictly smaller than $\alpha$, and if $B_1$, $B_2$ and $B_3$ are 'large enough' then it will be close to $\alpha$ and still smaller of equal to it.
>
> **Q5:** The proof of asymptotic level of the aggregated test with wild bootstrap relies on the asymptotic level of the single tests. This is proved by Chwialkowski et al., 2016, Proposition 3.2 in a mathematically precise way for the wild bootstrap: they prove that the difference between true quantiles and the wild bootstrap quantiles converges to zero in probability under the null hypothesis with the following dependence on $N$:
> $$
> \text{sup}_x \mid P(N B_N > x \mid Z_1, \dots, Z_N) - P(N K_N > x \mid Z_1, \dots, Z_N) \mid
> $$
> converges to 0 in probability, where $K_N$ is the KSD estimator of Equation (1) in our paper, and $B_N$ is the wild bootstrap KSD of Equation (4).
>
> **Q6:** In settings where the median bandwidth is the 'best' bandwidth, KSD median would be more competitive than KSDAgg since by considering a large collection of bandwidths we are not only considering the 'best' median bandwidth but also 'worse' bandwidths. However, in practice, we cannot know in advance which bandwidth would perform well, and KSDAgg retains power even for large collections of bandwidths (21 bandwidths considered in MNIST Normalizing Flow experiment). We could imagine a setting where the best bandwidth lies in between two bandwidths of our collection and those two are 'bad' bandwidths, in which case a test which uses data splitting to select an 'optimal' bandwidth would be able to select it, however one must bear in in mind the loss of power due to data splitting. In our experiements, the aggregation approach outperformed competing approaches.
>
> **Q7:** As presented in the inputs of Algorithm 1, it is sufficient for KSDAgg to have access to the score function (gradient of log density), this allows to work with unnormalized densities, as it is the case with Gaussian-Bernoulli Restricted Boltzmann Machine (Section 4.4) for example. We will emphasize this fact in the introduction. There are no extra assumptions for the wild bootstrap, while for the parametric bootstrap we need to have access to a sampler of the density as well, to the best of our knowledge it is not enough for this sampler to be approximate.
>
> **Q8:** When the squared $L^2$-distance of the difference in densities is smaller than the lower bound provided for all $\beta\in(0,1)$, this means that $p$ and $q$ are very close to each other and cannot be distinguished from each other at this fixed sample size. In this setting, we cannot provide any power guarantees (even deteriorated).
>
> **proof Theorem 3.3**: The proof Theorem 3.3 relies on (i) upper bounding a probability of intersections of events by the minimum of the probabilities of each event, (ii) checking that the assumptions of Theorem 3.1 are satisfied for the adjusted levels, (iii) applying Theorem 3.1. This proof sketch will be included in the final version of the paper.

---

> > ### Comment · Reviewer_ys6t · 2022-08-06
> > **Second response**
> >
> > Thank you for your well-presented response. A couple of points:
> >
> > 2. It will be good to state that the bounds for sample split are unknown, rather than claiming that they are guaranteed to be worse (at least that's the impression I currently get from your paper).
> >
> > 3. I am not sure I fully follow your comment---will a concrete example be mathematically presented for when the conditions in Theorem 3.1 / 3.3 are satisfied, e.g., given a certain mean separation in d-dimensions isotropic Gaussians? Or would just a numerical experiment be conducted? I did read your response to CDhe, and it seems that you are planning to only do experiments. If that is the case, it would be please clarify that it is non-trivial to derive any explicit theoretical result/conditions for your Theorems.
> >
> >
> > For all other comments, I am assuming you would add clarifications, and make statements more precise (e.g., 5) in your revision.

---

> > > ### Author Response · Authors · 2022-08-09
> > > **Response to Reviewer ys6t**
> > >
> > > We thank reviewer ys6t for suggestion 2, we will clarify this in the final version of the paper.
> > >
> > > Regarding comment 3, we point the reviewer to our comment **For all reviewers: Updated version of the paper (Appendix D)** and to the updated paper which now mathematically presents concrete examples for which the conditions in Theorem 3.1 / 3.3 are satisfied, and derives minimax optimal rates over Sobolev balls. Our results hold for any strictly positive compactly supported densities (with continuous score function for the model): this includes for example d-dimensional isotropic Gaussians truncated to some compact subset of $\mathbb{R}^d$. This makes precise the comment we had provided in lines 175--176.

---

### Author Response · Authors · 2022-08-02
**Comments to all reviewers**

We warmly thank all reviewers for their careful reading of our paper and their invaluable insights.
We hope this rebuttal addresses the concerns expressed by reviewers, and if so, that they would kindly consider upgrading their evaluation.
We provide some general comments here about novelty of our paper and differences with prior work, and we individually answer the questions of the reviewers below.

We consider the novelty aspect of our work to be the proposal of a solution to the kernel selection problem for the KSD goodness-of-fit setting. KSD tests are widely used and cited; despite their appearance back in 2016, kernel selection had still never been done for these tests. Our contribution is in showing, both theoretically and experimentally, that the aggregation procedure works in this novel setting. Contrary to previous works on two-sample and independence testing, we have presented our theoretical results in the more general framework of kernel selection rather than bandwidth selection.

Applying the aggregation procedure to the goodness-of-fit setting is not trivial, we highlight some of the main differences with the two other testing frameworks. In our case, using a wild bootstrap does not result in a test with well-calibrated non-asymptotic level: the reasoning used for the MMD and HSIC breaks down because of the asymmetry of the KSD with respect to the two densities. In order to guarantee correct non-asymptotic level for our aggregated test, we propose to use a parametric bootstrap, a procedure unique to the goodness-of-fit framework. The lack of translation invariance of the Stein kernel introduces new challenging problems. For the expectation of the Stein kernel squared, i.e.
$$
E_q[h_{p,\lambda}(X,Y)^2],
$$
it is not possible to extract the bandwidth parameter $\lambda$ outside of the expectation as it is the case for the usual kernel $k_\lambda$ when using either the MMD or HSIC. Moreover, working with the integral transform $$(h_{p,k}\diamond\psi)(y) = \int_{\mathbb{R}^d} h_{p,k}(x,y) \psi(x) \,\mathrm{d} x$$ is more complicated since the operation $\diamond$ does not correspond to a simple convolution, as would be the case when working directly with a translation-invariant kernel.

We also emphasise that we have experimentally validated our proposed approach on benchmark problems, not only on synthetic datasets classically used in the literature but also on original data obtained using state-of-the-art generative models (i.e. Normalizing Flows). We provide publicly available code to allow practitioners to employ our method (please see the link in the paper, line 69).

---

### Author Response · Authors · 2022-08-09
**For all reviewers: Updated version of the paper (Appendix D)**

We thank all reviewers for the engaging discussions.  Reviewers asked for specific cases satisfying the conditions of Theorem 3.1 and 3.3. In response, we have achieved significantly stronger results than originally promised in the rebuttal text: we have proved minimax optimality and adaptivity results when assuming only compactness of the support of the densities and continuity of the model score function, which we believe is a strong addition to our theoretical results.

These new results are presented in Appendix D of the updated submission (this will be included in the main text for the final version of the paper).  Specifically: under the abovementioned assumption that the densities are compactly supported, we have been able to derive the uniform separation rate over Sobolev balls for the KSD and KSDAgg tests using a Gaussian kernel. The rate is minimax optimal for the KSD test using a bandwidth depending on the unknown smoothness parameter of the Sobolev ball. KSDAgg is adaptive to this unobserved smoothness parameter and achieves the minimax rate up to an iterated logarithmic factor. The proof is based on noting that, in the compactly supported setting, the Stein kernel can be upper bounded by a Gaussian kernel which is translation-invariant. For a translation-invariant kernel, the integral transform is a convolution, which allows working in the Fourier domain through Plancherel’s Theorem, hence our choice of Sobolev balls in characterising the separation rate. We remark that the proof does not follow directly from the earlier result  'Schrab et al., 2021, MMD Aggregated Two-Sample Test': an additional term (L2-norm of the integral transform (convolution) of the difference in densities) needs to be dealt with.

---

### Meta-Review · Area_Chair_UAcW · 2022-08-26

**Recommendation:** Accept
**Confidence:** Certain

**Metareview:**

The paper proposes a novel method of statistical tests with Kernel Stein Discrepancy, aggregating multiple tests with different kernels.  The method can avoid data splitting, which is commonly used to choose a kernel aiming at better power but may not be effective with a smaller sample size.  The paper gives theoretical analysis, and also experimental results outperforming other relevant methods.  The work gives solid theoretical and methodological advances in the field of kernel-based tests.  We think the work is worth being presented in NeurIPS.

**Award:**

No

---

### Decision · Program_Chairs · 2022-09-14

Accept